# Therapeutic Potential of Astaxanthin for Body Weight Regulation: A Systematic Review and Meta-Analysis with Dose–Response Assessment

**DOI:** 10.3390/ph18101482

**Published:** 2025-10-01

**Authors:** Lucas Fornari Laurindo, Victória Dogani Rodrigues, Mauro Audi, Tereza Lais Menegucci Zutin, Mayara Longui Cabrini, Cláudio José Rubira, Cristiano Machado Galhardi, Jesselina Francisco dos Santos Haber, Lidiane Indiani, Maria Angélica Miglino, Vitor Engrácia Valenti, Eduardo Federighi Baisi Chagas, Sandra Maria Barbalho

**Affiliations:** 1Postgraduate Program in Structural and Functional Interactions in Rehabilitation, School of Medicine, Universidade de Marília (UNIMAR), Marília 17525-902, SP, Brazil; 2Laboratory for Systematic Investigations of Diseases, Department of Biochemistry and Pharmacology, School of Medicine, Universidade de Marília (UNIMAR), Marília 17525-902, SP, Brazil; 3Department of Biochemistry and Pharmacology, School of Medicine, Faculdade de Medicina de Marília (FAMEMA), Marília 17519-030, SP, Brazil; 4Department of Physiotherapy, School of Physiotherapy, Universidade de Marília (UNIMAR), Marília 17525-902, SP, Brazil; 5Department of Surgery, UNIMAR Charitable Hospital, Universidade de Marília (UNIMAR), Marília 17525-902, SP, Brazil; 6Department of Medicine, School of Medicine, Universidade de Marília (UNIMAR), Marília 17525-902, SP, Brazil; 7Department of Cardiovascular and Metabolic Health, School of Philosophy and Sciences, Universidade Estadual Paulista (UNESP), Marília 17525-900, SP, Brazil; 8Department of Research, Research Coordination Center, UNIMAR Charitable Hospital, Universidade de Marília (UNIMAR), Marília 17525-902, SP, Brazil

**Keywords:** carotenoid, body mass index, body weight, weight loss, dietary supplements, anti-obesity agents

## Abstract

**Background/Objectives:** Astaxanthin, a naturally occurring carotenoid renowned for its potent antioxidant properties, has been proposed as a dietary supplement for weight management due to its potential effects on adipose tissue and skeletal muscle metabolism, as well as its anti-inflammatory properties. This meta-analysis systematically evaluated the impact of astaxanthin supplementation on body mass index (BMI) and body weight in adult populations. **Methods:** Comprehensive searches of reputable databases were conducted, adhering to the PRISMA guidelines, with statistical analyses performed using Jamovi. **Results:** The study incorporated data from nine clinical trials. Pooled results indicated no significant reduction in the context of BMI (−0.2162; 95% CI: −0.4697 to 0.0374) and a non-significant decrease in body weight (0.0230; 95% CI: −0.4534 to 0.4994) relative to control groups. The heterogeneity observed across studies was 30.1251% (*p* = 0.1593) for BMI and 73.3885% (*p* = 0.0002) for body weight management. The dose–response analysis showed no statistically significant association between astaxanthin dosage and outcomes related to BMI and body weight management. Additionally, statistical assessment of funnel plot asymmetry indicated no evidence of publication bias. **Conclusions:** The findings indicate that astaxanthin does not provide benefits in BMI regulation nor in weight control management, highlighting the need for additional large-scale and long-term clinical trials. This study contributes to the growing body of evidence on the role of nutraceuticals in metabolic health, providing a foundation for future clinical recommendations.

## 1. Introduction

Obesity is a chronic, systemic, and progressive disease with growing prevalence worldwide. In the United States (US), it is estimated that over 40% of US adults and nearly 20% of US children have obesity. However, this number is expected to grow, since obesity is characterized as a relapsing and progressive condition and requires long-term, multi-faceted treatment strategies. Current evidence supports that several treatment strategies against obesity, including lifestyle interventions, medications, and bariatric surgery, are of the utmost importance. These improve health outcomes for people with obesity. However, more than treatment strategies, risk-reduction approaches are of the utmost importance, as they are complication-focused and risk-reduction alternatives targeting not only weight loss but also weight-related health issues [1].

Clinical obesity is diagnosed when there is evidence of organ or tissue dysfunction caused by an excessive amount of body fat or in patients with a very high body mass index (BMI greater than 40 kg/m^2^) and excess adiposity presenting with limitations to living due to significant body weight. According to the latest guidelines, people with clinical obesity should receive timely, evidence-based treatment to improve or reverse complications associated with their condition, and people with preclinical obesity, those who have excessive adiposity but without organ or life dysfunctions, should receive counseling, monitoring, and targeted interventions to reduce obesity-related risk [2,3]. In this scenario, bioactive compounds are crucial, as they modulate adipose tissue functions and can influence obesity and its associated complications. Natural compounds can regulate obesity-related inflammation and oxidative stress, insulin resistance, lipogenesis, adipogenesis, M1 macrophage recruitment, and energy production and expenditure. Therefore, these compounds can help shift adipose tissue function from an obese/inflamed state to a healthier, regulated one [4].

Astaxanthin (Figure 1) is a red-colored xanthophyll carotenoid, also known as the “king of carotenoids.” It is naturally found in green algae, red yeasts, and aquatic species, including salmon, krill, trout, and fish eggs [5]. Astaxanthin exhibits potent antioxidant [6] and anti-inflammatory effects [7], in addition to its anti-cancer [8], anti-diabetic [9], anti-aging [10,11], hepatoprotective [12], and cardiovascular protective properties [13,14,15]. Although it possesses low water solubility [16], astaxanthin is utilized in the food [17], agriculture [18], aquaculture [19], and cosmetic industries [20]. In addition to the effects mentioned earlier, astaxanthin exhibits anti-obesity properties, which are not as well-studied as the other health benefits associated with its structure. Evidence suggests that astaxanthin can modulate critical aspects related to visceral fat accumulation, adipose tissue lipid metabolism, and dyslipidemia [21]. It also exhibits anti-obesity effects in other key organs involved in lipid metabolism, including skeletal muscles. Within skeletal myotubes, astaxanthin improves lipid-associated lipotoxicity, restores impaired lipid metabolism, and improves oxidative damage related to the excess of lipids. In skeletal muscle cells, astaxanthin also modulates intramuscular fat accumulation, restricting lipid storage in muscles and protecting against adiposity-associated muscle dysfunction via critical molecular pathways [22].

Astaxanthin is principally derived from algae, bacteria, and yeast. However, it can also come from fish and shellfish when these animals eat microscopic marine algae or phytoplankton, which are the base of several aquatic food webs. Yeasts and bacteria often produce less astaxanthin than algae. Therefore, microalgae are a promising source of astaxanthin for use in livestock and poultry production. A notable example of algae is *Haematococcus pluvialis*, which contains 55 mg/g of dry weight astaxanthin, offering not only an excellent astaxanthin content but also high productivity. *H. pluvialis* biofilms are expected to reach an astaxanthin production of up to 204 mg/m^2^/day. *Phaffia rhodozyma*, a red yeast, produces up to 6.4 mg/g of dry weight of astaxanthin. *Escherichia coli*, a bacterium, produces up to 6.2 mg/g of dry weight of astaxanthin. Fish and shellfish contain up to 4 mg/kg and 2.1 g/kg of astaxanthin, respectively [30,31,32]. Thus, while yeast, bacteria, fish, and shellfish contribute to astaxanthin availability, microalgae, especially *H. pluvialis*, remain the most promising and productive natural source for large-scale production.

Therefore, our main objective was to conduct a systematic review of randomized clinical trials and a meta-analysis of astaxanthin supplementation on BMI and body weight management. Our approach was also enriched with a dose–response assessment and a review of astaxanthin’s pharmacological properties, including its molecular characteristics, pharmacokinetics, and pharmacodynamics associated with its anti-obesity effects. To the best of our knowledge, this is the first review to fully elucidate the benefits of astaxanthin in controlling weight. Before this review, two meta-analyses assessed the effect of astaxanthin supplementation on obesity-related disorders. Firstly, Radice et al. [33] found that astaxanthin supplementation was beneficial for symptoms associated with obesity-related diseases. However, their analysis did not focus on weight control, and the included studies were solely preclinical animal trials. Secondly, Xia et al. [34] conducted a systematic review and meta-analysis to study the effects of astaxanthin supplementation on obesity, blood pressure, inflammatory biomarkers, glycemic control, and lipid profile in human subjects. However, their analysis was flawed in the face of novel discoveries, as since 2020 many new publications and clinical trials have been published on astaxanthin supplementation and obesity-related disorders, as well as astaxanthin’s pharmacokinetics and pharmacodynamics. In addition, the present meta-analysis yielded results from the included studies to run a dose–response meta-analysis, highlighting how controlling astaxanthin dosages could influence final results regarding BMI or weight management. Furthermore, our methodology was based on the reported PICOS (Population, Intervention, Comparison, Outcomes, and Study Design) framework, which significantly enhances the validity and transparency of our findings due to enhanced reproducibility. Finally, the focus on weight management and BMI control was previously insufficient, as their approach was overly general and applicable to a range of metabolic risks and outcomes. The current systematic review and meta-analysis aims to fill these gaps in the current literature.

## 2. Materials and Methods

### 2.1. Focus Question

The present systematic review and meta-analysis with dose–response assessment were designed to answer the question, “*What are the effects of astaxanthin in improving the body mass index and body weight of human subjects?*”.

### 2.2. Language

This systematic review and meta-analysis included solely clinical trials published initially in the English language, which did not introduce systematic bias, as a previously published review paper concluded that there was no evidence of a reported systematic bias in literature reviews due to English language restrictions [35].

### 2.3. Literature Search and Databases

To conduct this systematic review and meta-analysis, we searched reputable databases, including PubMed, SpringerLink, ScienceDirect, Cochrane, Google Scholar, Scopus, and Web of Science. Mesh terms such as “Astaxanthin,” “Body Mass Index,” “Body Weight,” “Obesity,” “Anti-Obesity Effects,” “Signaling Pathways,” and “Clinical Trials” were utilized to ensure rigor and consistency in our search. Boolean operators “AND” or “OR” were utilized. Their use refined the search process, yielding better results. All retrieved articles were exported to the Rayyan QCRI program (Qatar Computing Research Institute, Qatar, Online application) to eliminate duplicates. Rayyan application screened the retrieved studies by assessing their titles and abstracts. After this stage, L.F.L. and V.E.V. completed the suitability stage by reading the manuscripts selected after duplication screening. Another author (S.M.B.), besides deciding on the occurrence of a disagreement between the two initial reviewers, decided whether specific studies should be included. At the final stage of the literature search across all reputable databases, the authors evaluated the possibility of conducting a meta-analysis within the group after assessing all of the included studies (that met the eligibility criteria). There is no registration number for this systematic review and meta-analysis.

### 2.4. Study Selection

All studies included in this review were sourced from reputable and peer-reviewed journals. These publications spanned from the inception of the database to July 2025. The inclusion and exclusion criteria were guided by the PICOS framework, which encompasses the elements of PICOS:

(P) Patients older than 18 years old.

(I) The intervention group ought to receive astaxanthin as an intervention.

(C) Patients who received a placebo for the effect of comparisons between groups.

(O) Body mass index and body weight.

(S) We included randomized and placebo-controlled trials. We incorporated only articles sourced from reputable, peer-reviewed journals and authored in English. The following categories were excluded from consideration: conference papers, abstracts, master’s dissertations, doctoral theses, descriptive studies, commentaries, case studies, editorials, and reviews. Nevertheless, prior reviews regarding the broad effects of astaxanthin were employed to inform specific sections of our current manuscript, including the Introduction. The included studies spanned up to July 2025.

### 2.5. Data Extraction

All relevant data, including authorship, publication date, study design, number of included participants (treated or untreated with the intervention), prognostic and demographic information, and data on the intervention or control protocols of the respective studies, were included. This information was extracted from primary studies and presented in a table. All important missing data were requested by contacting the corresponding author of each respective study. Two reviewers (E.F.B.C. and V.D.R.) finished this stage independently. Data extraction from published graphs was conducted using Web Plot Digitizer^®^ in instances where the corresponding authors did not provide a response. The lipid profile data were represented as means with corresponding standard deviations. Instances where the primary studies reported “standard error” or “confidence intervals” were converted to standard deviations for the purpose of data consistency.

### 2.6. Search and Selection of Relevant Articles

We followed the Preferred Reporting Items for Systematic Reviews and Meta-Analyses (PRISMA) guidelines, as reported by Page et al., to collect and retrieve all relevant data from the included studies. The writing of this manuscript’s text also corresponded with the PRISMA checklist for main text and abstracts, as reported in PRISMA’s publication [36].

### 2.7. Data Items

We collected the means and standard deviations of BMI and body weight. Additionally, to minimize the risk of bias in our systematic review and meta-analysis, data on participant and intervention profiles, funding sources, and support across the included articles were obtained from the selected references. At this point, missing or unclear information was discarded.

### 2.8. Quality Assessment

The authors utilized the COCHRANE Handbook for Intervention Assessments [37] to evaluate the bias of the included studies. This document outlines critical aspects to consider in the research methodology, including the focus of the research question, the implementation of appropriate randomization techniques, and the establishment of allocation blinding. It also addresses participant attrition, the collection of prognostic and demographic characteristics, the assessment of outcomes, the application of intention-to-treat analysis, the execution of sample size calculations, and the importance of ensuring sufficient follow-up duration.

### 2.9. Qualitative Analysis

We conducted a narrative synthesis to provide a comprehensive description of the completion of each study. The details of each study were presented in both textual and tabular formats. The results of the surveys concerning BMI and body weight intervention with astaxanthin supplementation were determined by comparing the outcomes between intervention and control (placebo) groups.

### 2.10. Synthesis of Results and Summary Measures

The statistical analysis was performed using Jamovi open statistics software (Version 2.6.26, Solid). A *p*-value of less than 0.05 was established as the criterion for determining statistical significance. In conducting this meta-analysis, the standardized mean difference was utilized as the outcome measure, and a random-effects model was applied to the dataset [38]. Heterogeneity, quantified as tau^2^, was assessed utilizing the DerSimonian–Laird estimator. In addition, we reported the Q-test for heterogeneity alongside the I^2^ statistic. In instances where heterogeneity was identified (i.e., tau^2^ > 0, independent of the Q-test results), we projected the interval for the actual outcomes. Furthermore, we employed studentized residuals and Cook’s distances to investigate the potential presence of outliers and/or influential studies within the context of the model. In the context of this study, a studentized residual exceeding the 100 × (1 − 0.05/(2 × k))th (where k is the number of included results) percentile was identified as a potential outlier, indicating deviations from a standard normal distribution. This assessment employed the Bonferroni correction test, applying a two-sided significance level of α = 0.05 for the studies included in this meta-analysis. Furthermore, studies were classified as influential when their Cook’s distance exceeded the median plus six times the interquartile range of the Cook’s distances. To evaluate the asymmetry of the funnel plots, the correlation test, regression test ranks, and the standard error of the observed outcomes were utilized as predictors [39,40]. For the dose–response meta-analysis, the continuous moderator (astaxanthin’s dose) was included in the assessment.

## 3. Results

A total of 258 records were identified from the databases searched during the initial identification of the included studies (100 studies from PubMed, 42 studies from SpringerLink, 23 studies from ScienceDirect, 10 studies from Cochrane, 10 studies from Google Scholar, 25 studies from Scopus, and 48 studies from Web of Science). Among them, 142 records were removed from the sample due to duplication, 59 were identified as ineligible by automation tools (Covidence), and 40 were excluded for other reasons. These reasons included inconsistencies in the methodology that conflicted with our selection criteria at this stage. Following this stage, 17 records were screened, and 7 additional records were excluded. The reason for excluding these 7 reports was that they lacked involvement of astaxanthin as an intervention. After this, 10 records were sought for retrieval. Fortunately, all reports were successfully retrieved. Finally, 1 report was excluded due to a lack of astaxanthin involvement as an intervention. We included 9 studies in the final analysis. Figure 2 presents a flow diagram that outlines the literature search process and the studies included, following the PRISMA guidelines. Some studies appear more than once in the PRISMA flow diagram because they assessed different doses of astaxanthin and were therefore considered in multiple independent comparisons. Table 1 outlines the studies included following the PRISMA guidelines. Table 2 provides a bias assessment for each study included, as outlined in the Cochrane Handbook.

## 4. Discussion

### 4.1. Pharmacokinetics of Astaxanthin: Its Absorption, Distribution, Metabolism, and Excretion

Carotenoids such as astaxanthin are absorbed into the body like lipids [50]. Then, they are transported by the lymphatic system into the liver. Since astaxanthin is a fat-soluble compound, its absorption depends on accompanying dietary components. In this context, astaxanthin is more easily absorbed with lipids, such as cholesterol and phospholipids, and emulsified bile salts, forming lipid digestion products that accumulate to form micelles. Micelle formation controls lipolysis reactions, which are required for lipid transportation. This transport occurs from the duodenal lumen to the intestinal water layer, which comprises the brush border of the epithelial cells lining the duodenum (enterocytes), the location where absorption occurs. Once the digestion of lipids is complete, the products, which are mainly fatty acids and monoglycerides or glycerol, are solubilized by bile salts, allowing the formation of mixed micelles that are transported to the enterocytes for further processing. Enterocytes then secrete these micelles into the lymphatic system. These steps are preceded by the release of astaxanthin from the food matrix and its dissolution by various digestive enzymes associated with lipid digestion. Astaxanthin can be freely released in the gastrointestinal tract and penetrate enterocytes through selective mediation by membrane transport proteins. However, the most common form of astaxanthin transportation is through the bloodstream via low-density lipoprotein (LDL) and high-density lipoprotein (HDL) particles [51,52,53]. It is well established that different isomers of astaxanthin are absorbed at varying rates, with the (3S, 3 ′S) isomer exhibiting the highest plasma levels. The carotenoid exhibits a non-linear dose–response relationship and selective absorption of Z-isomers. Its half-life is about 52 h, and astaxanthin molecules from aquaculture fish appear to be more bioavailable than those from wild fish [54]. Therefore, source and isomer composition influence the absorption and effectiveness of astaxanthin in the human body. The liver primarily metabolizes astaxanthin, and the kidneys are responsible for its excretion, following the spleen. The spleen, kidney, heart, lung, and liver, respectively, have the highest total concentrations of astaxanthin reserves in the body [55,56,57]. Recommended or approved doses for astaxanthin vary from 2 to 24 mg a day. A review of 87 human studies assessing the safety of astaxanthin concluded that there are no safety concerns associated with natural astaxanthin supplementation, with the majority of the included studies having doses above 12 mg daily [58].

### 4.2. Exploring the Differential Anti-Obesity Effects of Astaxanthin: Emphasizing Its Effects on Adipocytes from the Adipose Tissue and Myotubes from the Skeletal Muscles

Although astaxanthin presents several antioxidant, anti-inflammatory, cardiovascular protective, and cerebrovascular protective effects, its effects against obesity are less well understood. The effectiveness of a bioactive compound against obesity can be understood by examining its roles in adaptogenic differentiation and adipocyte function. Free astaxanthin has no effects on adipocyte proliferation at low concentrations. However, even at low concentrations, astaxanthin can inhibit adipocyte differentiation by suppressing the expression of peroxisome proliferator-activated receptor gamma (PPARγ), CCAAT/enhancer-binding protein alpha (C/EBPα), and sterol regulatory element-binding protein-1c (SREBP-1c), while upregulating the expression of Wnt10b, low-density lipoprotein receptor-related protein 6 (LRP6), and frizzled (FZ) in the Wnt/β-catenin signaling pathway and increasing β-catenin translocation into the nucleus in adipocytes [59]. These findings suggest that the carotenoid astaxanthin may exert anti-adipogenic effects by modulating key transcription factors and activating the Wnt/β-catenin signaling pathway, thereby inhibiting adipocyte differentiation without affecting cell proliferation. Astaxanthin’s inhibition of PPARγ has also been associated with the downregulation of messenger ribonucleic acids (mRNA) of key genes related to lipogenesis inhibition, including acetyl-CoA carboxylase (ACC), fatty acid synthase (FAS), adipocyte fatty acid-binding protein (aP2), cluster of differentiation 36 (CD36), and lipoprotein lipase (LPL) in 3T3-L1 adipocytes [60,61]. This suggests that astaxanthin’s inhibition of PPARγ can not only suppress adipocyte differentiation but also downregulate lipogenesis by suppressing the expression of key lipogenic genes, inhibiting both fatty acid and triglyceride synthesis within adipocytes and, therefore, exerting anti-lipogenic effects in 3T3-L1 cells.

These interactions have been confirmed in high-fat diet-fed beagles, as astaxanthin-supplemented animals exhibited a more nuanced weight loss period, with reduced activation of lipid synthesis-related pathways and increased activation of energy metabolism-related pathways [62]. In obese mice, astaxanthin also regulates adipose tissue inflammation and fibrosis, significantly reducing pro-inflammatory cytokine levels, decreasing M1 macrophage infiltration and the expression of pro-inflammatory macrophage markers, and inhibiting the expression of hypoxia-inducible factor 1 alpha (HIF-1α) and its downstream fibrogenic genes. These anti-fibrotic and anti-inflammatory effects have also been confirmed in obese mouse livers [63]. This suggests that, in the context of obesity, astaxanthin also exerts anti-inflammatory and anti-fibrotic effects in adipose tissue by reducing levels of pro-inflammatory cytokines, limiting M1 macrophage infiltration, and suppressing HIF-1α signaling, which directly contributes to the improvement of adipose tissue function and metabolic health.

Another anti-obesity pathway induced by astaxanthin is related to skeletal muscles. Astaxanthin stimulates the mitochondrial fatty acid oxidation capacity of muscular cells in obese mice, thereby regulating mitochondrial biogenesis in skeletal muscles and reducing lipid accumulation within adipose tissue cells [63]. This strongly suggests that astaxanthin enhances mitochondrial fatty acid oxidation and biogenesis in skeletal muscle cells, playing a crucial role in systemic lipid metabolism and ultimately contributing to reduced fat storage and improved energy balance in obesity settings.

In skeletal muscles, astaxanthin regulates mitochondrial biogenesis principally through adenosine monophosphate-activated protein kinase (AMPK) activation in myotubes. This effect can also be beneficial in increasing skeletal muscles’ exercise tolerance and also exerting anti-inflammatory effects via antioxidant mechanisms [64]. These effects further reinforce astaxanthin’s therapeutic potential in managing obesity and its related metabolic disorders, as obesity is not only associated with fat storage [65]. Indeed, it involves systemic metabolic dysfunction, including inflammation, insulin resistance, and reduced energy expenditure. By promoting mitochondrial biogenesis in skeletal muscle cells, astaxanthin restores energy metabolism and glucose homeostasis. Through its anti-inflammatory and antioxidant effects, astaxanthin reduces the chronic, low-grade inflammation characteristic of obesity. Ultimately, by enhancing exercise tolerance, astaxanthin promotes physical activity, a cornerstone of effective obesity management.

Astaxanthin is also associated with improved insulin resistance. In L6 cells (myotubes), astaxanthin positively regulates glucose transporter type 4 (GLUT4) translocation from the plasma to the membrane, enhancing protein kinase b (Akt) phosphorylation via feedback activation of the phosphatidylinositol 3-kinase (PI3K) signaling pathway in these cells [66]. These cellular findings strongly position astaxanthin as a potential enhancer of insulin sensitivity. Its effects in promoting GLUT4 translocation and Akt phosphorylation through the PI3K pathway in myotubes are sufficient for astaxanthin to restore insulin responsiveness in skeletal muscle cells, thereby maintaining glucose homeostasis, highlighting its possible role in modulating insulin resistance associated with obesity. Astaxanthin can also reduce lipotoxicity and skeletal muscle dysfunction during obesity by modulating the gut microbiota, thereby enhancing the abundances of *Akkermansia*, *Bifidobacterium*, *Butyricicoccus*, and *Staphylococcus* strains in mice fed a high-fat diet [67]. By reshaping the gut microbiome, astaxanthin helps to alleviate obesity-induced muscle dysfunction and lipotoxicity, highlighting the gut–muscle axis as a key therapeutic target of this carotenoid’s effects against obesity and obesity-related disorders, offering a novel approach to mitigate these metabolic disturbances. Figure 3 illustrates the main effects of astaxanthin against obesity.

### 4.3. Overview of the Included Studies and Quality and Bias Assessment

In a study conducted with women diagnosed with polycystic ovary syndrome, Jabarpour et al. [41] evaluated the effects of 12 mg of astaxanthin administered daily (*n* = 26; age, 30.42 ± 4.69 years; BMI, 26.08 ± 1.89 kg/m^2^) in a randomized controlled (*n* = 27; age of 31.19 ± 4.57 years; BMI of 26.55 ± 1.89 kg/m^2^) clinical trial for eight weeks. Their trial revealed no significant effects of astaxanthin on BMI. No adverse effects were reported. However, the study had some limitations that must be addressed. Firstly, the small sample size, the brief intervention duration, and the lack of serum astaxanthin concentration assessments are considered severe limitations. Secondly, the study lacked appropriate randomization generated by computer-based techniques. Additionally, the results lacked an intention-to-treat analysis, which could limit the generalizability of the findings due to the exclusion of data from participants who dropped out before the study was completed [41].

Saeidi et al. [42] conducted a randomized controlled trial to evaluate the effects of astaxanthin supplementation in 68 men with obesity. The intervention consisted of 20 mg of astaxanthin administered daily for 12 weeks (*n* = 17, body weight of 94.2 ± 2.6 kg, fat percentage of 31.1 ± 1.5%, BMI of 33.2 ± 1.4 kg/m^2^) or a placebo (*n* = 17, body weight of 95.3 ± 1.8 kg, fat percentage of 31.1 ± 1.5%, BMI of 34.1 ± 2.5 kg/m^2^). The results of this study showed that astaxanthin was effective in reducing body weight among participants, with a statistically significant result (*p* = 0.008). The same did not occur in the placebo group. The participants’ BMI was also significantly decreased in the astaxanthin group (*p* = 0.019), but not in the placebo group. Astaxanthin also considerably reduced body fat (*p* = 0.004) compared to the placebo group, which showed no positive effects on body fat. Although this study had its strengths, which are discussed above, it also had some limitations that must be addressed. Firstly, there was a high dropout rate, and the study lacked sufficient evidence that the authors employed an intention-to-treat approach. Therefore, the generalizability of their findings may be in question, as these exclusions could have altered the results. Additionally, the randomization process of the participants was not specified, nor was it mentioned whether it was implemented using computer-based techniques, which could raise concerns about selection bias. Finally, there was no allocation blinding, and insufficient information was provided on prognostic and demographic characteristics, which strengthens the concerns about selection bias in this study.

In a study conducted in the US, Liu et al. evaluated the effects of astaxanthin in healthy older adults compared with a placebo. For the intervention group (*n* = 9 men, age 69.2 ± 1.0 years, body weight ~83.55 ± 2.27 kg), astaxanthin was administered at a dose of 12 mg as two capsules per day for 12 weeks. In this study, astaxanthin did not demonstrate statistical significance in decreasing body weight compared to the placebo. The study also had some limitations that must be addressed. Firstly, the intervention group consisted of a small sample size. Secondly, the authors did not evaluate the biochemical mechanisms underlying astaxanthin supplementation on the utilization of fat and carbohydrates. Finally, despite its small sample size and significant losses (>20%), the study’s findings were not verified using the intention-to-treat analysis approach among all study participants, including those who discontinued the intervention [43].

In another study conducted by Mashhadi et al. [44], 44 participants with type 2 diabetes mellitus were randomly assigned to receive 8 mg of astaxanthin daily for 8 weeks (8 ♂ and 14 ♀, age of 51 ± 9.7 years, BMI of 30.0 ± 5.11 kg/m^2^, total body fat of 35.5 ± 10.6%) or placebo (9 ♂ and 13 ♀, age of 54 ± 8 years, BMI of 30.4 ± 5 kg/m^2^, total body fat of 39.7 ± 9.8%). There were no discernible differences in BMI or total body fat between the groups after the study, and the trial was unable to determine the molecular mechanisms associated with astaxanthin’s effects. The limitations included a lack of methodological rigor in randomizing the included participants, inadequate reporting of sample calculation, and failure to utilize an intention-to-treat analysis approach to determine the most generalizable effects based on the entire sample, including those who discontinued the intervention.

In Korean patients, Choi et al. [45] investigated the effects of astaxanthin at a dose of 20 mg daily for 12 weeks (12 ♂ and 2 ♀, age of 31.1 ± 9.4 years, BMI of 28.1 ± 2.4 kg/m^2^, body weight of 83.6 ± 9.4 kg, height of 1.72 ± 0.07 m) on overweight adults compared to a placebo (11 males and two females, age of 30.1 ± 9.5 years, BMI of 26.3 ± 1.3 kg/m^2^, body weight of 77.1 ± 10.8 kg, height of 1.71 ± 0.10 m). The results demonstrated that BMI and body weight were minimally changed in the astaxanthin group. However, the intervention yielded a few adverse events, including a change in fecal color to red, which was probably related to use of astaxanthin, and an increase in bowel movements. Although this study utilized 20 mg of astaxanthin, which is a higher dose of the carotenoid, the authors did not evaluate the effects of astaxanthin on antioxidant and anti-inflammatory parameters within the body, which may have been related to the minimal body weight and BMI alterations due to the treatment, as the control subjects experienced increased body weight and BMI values. Additionally, it appears that the authors did not employ appropriate randomization techniques in their study, as insufficient information was reported in this regard [45].

Within Japanese non-obese subjects, Yoshida et al. [46] evaluated the effects of a treatment with 6, 12, and 18 mg/day of astaxanthin on metabolic and non-metabolic parameters, including those related to obesity. The study comprised three intervention groups and one placebo group. Each intervention group received one of the three abovementioned astaxanthin treatment strategies. Although this study comprised a relatively small sample size, one considerable strength was the utilization of three different astaxanthin schemes to achieve different outcomes. However, body weight and BMI were unaffected at all doses. No adverse effects were reported even in the 18 mg/day astaxanthin group. Future studies with these different astaxanthin treatment schemes are needed to evaluate not only the potential impact of higher dosages with extended treatment durations on the body weight and BMI of human subjects but also to investigate the possible mechanisms underlying astaxanthin supplementation-related effects in humans.

Within Japanese patients, Nakagawa et al. [47] evaluated the effects of astaxanthin as a functional supplement for improving phospholipid peroxidation status in human erythrocytes using a randomized, double-blind, placebo-controlled design. Thirty health volunteers were recruited, and the intervention group consisted of twenty individuals equally divided into two astaxanthin dosage regimens (6 or 12 mg/day) over 12 weeks. The study did not yield statistical outcomes for changes in body weight or BMI, since weight management was not the primary outcome for the reported data. However, both the placebo and intervention groups experienced slight reductions in body weight and BMI after the intervention protocol, except for the BMI in the 12 mg/day astaxanthin group. Although this study evaluated the effects of astaxanthin in improving oxidative status in human cells, more critically designed studies must be conducted to evaluate the effectiveness of astaxanthin supplementation based on its antioxidant potential for anti-metabolic syndrome effects, such as improving body weight and lipid profile analyses.

Another study by Liu et al. [48] was conducted in the US with a randomized, double-blind, placebo-controlled design. At this time, 58 individuals experienced treatment regimens with astaxanthin 12 mg consumed as two capsules per day; however, the study presented a high dropout rate of 16 participants. As in the abovementioned study, this study did not report statistical analyses for changes in body weight and BMI measurements, since weight management was not the primary outcome analyzed for the reporting of data. However, at the study’s end, the placebo group experienced elevations in body weight and BMI, while the intervention group presented decreases in both parameters. One limitation of this study was that the authors did not report inflammatory or oxidative stress biomarkers by which aging-associated muscle decline might be explained. If astaxanthin helps to preserve or restore muscle strength and size, it may indirectly support weight management by maintaining a higher metabolic rate. In addition, inflammation and oxidative stress are linked to insulin sensitivity and fat accumulation [68,69,70].

Finally, Chen et al. [49] conducted a randomized, double-blind, placebo-controlled trial with 29 healthy Japanese women patients to evaluate whether astaxanthin could improve liver and leukocyte parameters. The intervention yielded a 12 mg/day astaxanthin regimen for 3 months. BMI changes were not statistically significant (*p* = 0.81), and the authors did not analyze body weight changes. Although this study indicated that astaxanthin supplementation may have hepatoprotective effects, its specific mechanisms in alleviating obesity-related liver dysfunction are still unclear. Future studies should elucidate the specific pathways affected by astaxanthin in liver cells, examining its effects on oxidative stress parameters, inflammation biomarkers, and lipid metabolism. This could lead to a more validated relationship between astaxanthin supplementation and body weight and BMI management under various metabolic health conditions.

Table 3 summarizes the key differences in the survey for stratified analysis of the risk of heterogeneity assessment. As demonstrated in this table, our included studies yielded different aspects regarding astaxanthin dosage, intervention duration, age, health status of the studied population, intervention protocol, and how the authors conducted their analyses based on the studied population. Astaxanthin dosage varied from 6 to 20 mg per day, while intervention durations typically spanned 8 to 12 weeks, with one extending to 4 months. Participant age varied widely, from young adults to older adults. The health status of participants also differed, including healthy individuals and those with conditions such as ovarian diseases, obesity, diabetes, and elevated triglycerides. Intervention protocols differed mainly in dosing frequency and timing. Subgroup analyses were primarily based on dosage levels, comparisons with placebo, and participant health status.

### 4.4. Report of the Quantitative Assessment Results: Assessing Astaxanthin’s Effects on Body Mass Index and Body Weight Through a Meta-Analysis of Randomized Controlled Studies

#### 4.4.1. Meta-Analysis for Body Mass Index

A total of 8 studies, comprising 11 results, were included in the analysis for BMI assessment. The observed standardized mean differences ranged from −0.9654 to 0.5289, with most estimates being negative (55%). The estimated average standardized mean difference based on the random-effects model was −0.2162 (95% CI: −0.4697 to 0.0374). Therefore, the average outcome did not differ significantly from zero (z = −1.6709, *p* = 0.0948). According to the Q-test, there was no significant amount of heterogeneity in the actual outcomes (Q_(10)_ = 14.3113, *p* = 0.1593, tau^2^ = 0.0548, I^2^ = 30.1251%). A 95% prediction interval for the actual outcomes was given by −0.7404 to 0.3081. Hence, although the average outcome was estimated to be negative, some studies may have revealed a positive outcome. An examination of the studentized residuals revealed that none of the studies had a value larger than ±2.8376, indicating no outliers in the context of this model. According to the Cook’s distances, none of the studies can be considered overly influential. Neither the rank correlation nor the regression test indicated any funnel plot asymmetry (*p* = 0.6481 and *p* = 0.7290, respectively). Figure 4 illustrates the results for BMI, along with the respective forest plot.

#### 4.4.2. Meta-Analysis for Body Weight

A total of six studies, comprising nine results, were included in the analysis for assessing body weight. The observed standardized mean differences ranged from −0.8867 to 1.3197, with most estimates being positive (56%). The estimated average standardized mean difference based on the random-effects model was 0.0230 (95% CI: −0.4534 to 0.4994). Therefore, the average outcome did not differ significantly from zero (z = 0.0947, *p* = 0.9246). According to the Q-test, the actual outcomes appeared to be heterogeneous (Q_(8)_ = 30.0622, *p* = 0.0002, tau^2^ = 0.3869, I^2^ = 73.3885%). A 95% prediction interval for the actual outcomes was given from −1.2859 to 1.3320. Hence, although the average outcome was estimated to be positive, some studies found negative outcomes. An examination of the studentized residuals revealed that none of the studies had a value larger than ±2.7729 and that there were no potential outliers in the context of this model. According to the Cook’s distances, none of the studies can be considered overly influential. Neither the rank correlation nor the regression test indicated any funnel plot asymmetry (*p* = 0.7614 and *p* = 0.7082, respectively). Figure 5 illustrates the results for body weight, along with the respective forest plot.

### 4.5. Dose–Response Meta-Analysis and Plot Asymmetry Analysis

Figure 6 illustrates the results for the dose–response meta-analysis for BMI and body weight assessments, along with the respective *p*-values. The dose–response meta-analysis revealed no differences in improvements in body weight or BMI with increasing dose, as indicated by the included results. For BMI, the addition of the moderator (dose assessment) did not improve control of the variable (BMI), indicating that the inclusion of the moderator had no statistical significance. The same occurred regarding body weight. Therefore, controlling the dose of astaxanthin supplementation did not alter the outcome measure. Thus, altering the dose of the supplement did not change the previously mentioned results.

Our funnel plot analysis revealed no asymmetry in the plots, indicating good reliability in the reported outcomes. Figure 7 illustrates the funnel plots for BMI and body weight assessments, along with the respective regression values.

## 5. Conclusions and Future Research Directions

This systematic review and meta-analysis demonstrated that astaxanthin did not significantly improve the BMI (−0.2162; 95% CI: −0.4697 to 0.0374). Also, astaxanthin did not yield a significant decrease in body weight (0.0230; 95% CI: −0.4534 to 0.4994). In addition, it is essential to note that the heterogeneity observed across studies was 30.1251% (*p* = 0.1593) for BMI and 73.3885% (*p* = 0.0002) for body weight.

Although this meta-analysis did not find positive effects regarding BMI and body weight management using the carotenoid astaxanthin, future research endeavors should still reveal a positive impact on overall weight management, including BMI control, due to the multifaceted anti-obesity benefits of astaxanthin. In this scenario, future studies should focus on subpopulations (e.g., individuals with metabolic syndrome, diabetes, hypertension, or other cardiovascular diseases) to assess the effects of this compound on health outcomes, taking into account the health status of the individuals being treated. Researchers must also attempt to elucidate the anti-obesity mechanisms of action of astaxanthin, incorporating nutrigenomics and genome-wide association studies into their arsenal. These approaches may reveal how astaxanthin influences gene expression, particularly concerning conditions such as obesity, which is closely linked to genetic predisposition and phenotypic interventions. Since astaxanthin presents solubility issues, assessing its potential synergistic effects with other lipid-based nutraceuticals may enhance its effectiveness in improving anti-obesity effects, including those related to muscles and adipocytes, as well as anti-inflammatory and antioxidant parameters. Combining astaxanthin with omega-3 fatty acids (e.g., Icosapent ethyl) or other plant sterols may be particularly important in this context.

In larger, well-designed clinical trials, researchers should also prioritize investigating different formulations of astaxanthin (e.g., free, natural, or synthetic) to assess the solubility and bioavailability of the various astaxanthins. These different types of astaxanthin may present differing dose-responsiveness, tolerability, pharmacoeconomics, and long-term cardiovascular outcomes. Therefore, one formulation may present improved anti-obesity effects compared to others, enhancing its effectiveness and necessitating a more nuanced approach to translational applicability in clinical practice. By exploring these research avenues, research on astaxanthin and its potential in combating obesity will evolve, and, in the meantime, the “king of carotenoids” may be effectively translated into clinical practice and introduced into clinical guidelines.

Our review presents a limitation despite its significant findings. While it highlighted essential results related to BMI and body weight management using the carotenoid astaxanthin, it was based solely on nine randomized clinical studies included in the final analysis. This limited number of studies restricts the generalizability of the findings, as greater statistical power is achieved by including a larger number of relevant studies.

## Figures and Tables

**Figure 1 pharmaceuticals-18-01482-f001:**
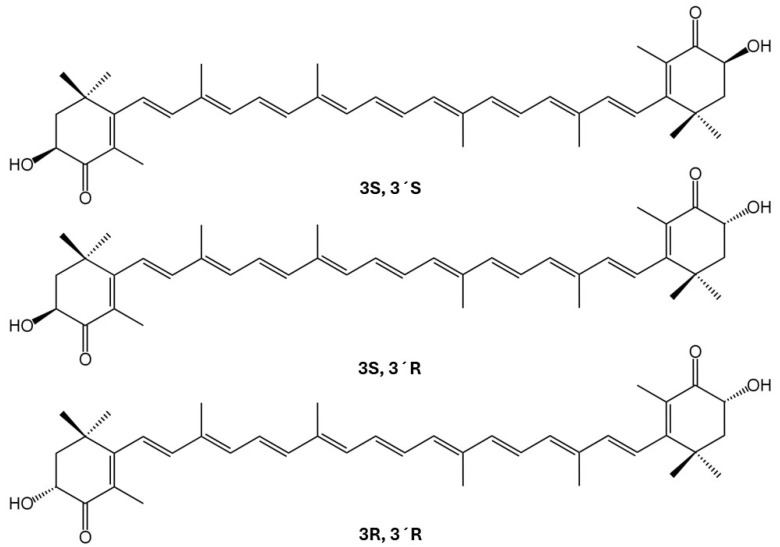
The isomeric forms of astaxanthin, a carotenoid [23]. It contains carbon, hydrogen, and oxygen atoms in its structure, consisting of two terminal rings joined by a polyene chain. In astaxanthin, oxygen atoms are present as OH and oxi groups. However, it is the polyene chain that gives this carotenoid its unique molecular structure. Astaxanthin can be found esterified in one or both hydroxyl groups with various fatty acids, including palmitic, oleic, stearic, and linoleic acids. However, this carotenoid can also be found free from hydroxyl group esterification [24,25]. According to PubChem, astaxanthin comprises the formula C_40_H_52_O_4_, has a molecular weight of 596.8 g/mol, and has an XLogP3-AA of 10.3 (meaning that the molecule possesses very high lipophilicity). Astaxanthin has a rotatable bond count of ten, which can impact its binding and bioavailability. Its charge is zero. However, its complexity score is 1340, indicating a more complex structure, having nine defined bond stereocenters [26]. However, these limitations can be overcome by microencapsulation and microemulsion using eutectic solvents or even nanoparticles, such as those made from phytoglycogen [27,28]. Astaxanthin possesses strong antioxidant and anti-inflammatory properties [29].

**Figure 2 pharmaceuticals-18-01482-f002:**
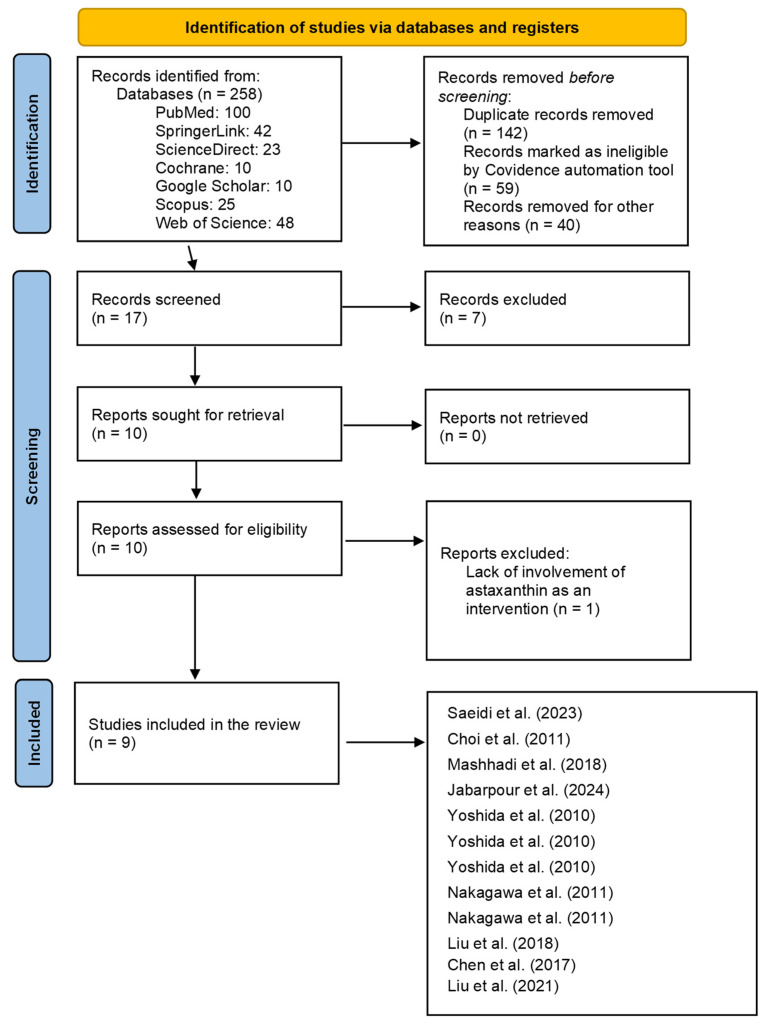
Flow diagram depicting the literature search process following the PRISMA guidelines [36,41,42,43,44,45,46,47,48,49].

**Figure 3 pharmaceuticals-18-01482-f003:**
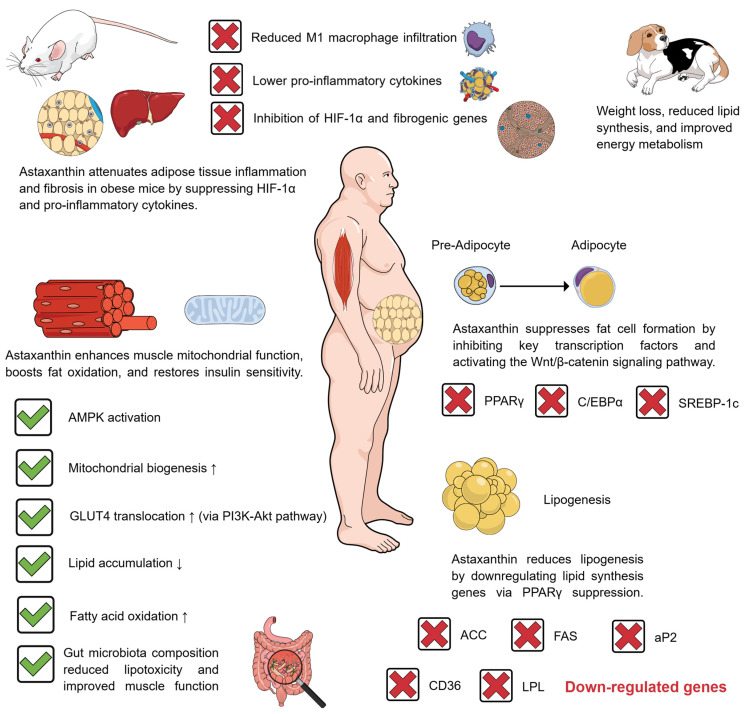
Astaxanthin attenuates obesity through multi-pathway modulation of adipose tissue, skeletal muscle, and the gut microbiota. Created using Mind the Graph (https://mindthegraph.com/), accessed on 18 September 2025. ↑: increase; ↓: decrease; ACC: acetyl-CoA carboxylase; Akt: protein kinase b; AMPK: adenosine monophosphate-activated protein kinase; aP2: adipocyte fatty acid-binding protein; CD36: cluster of differentiation 36; C/EBPα: CCAAT/enhancer-binding protein alpha; FAS: fatty acid synthase; GLUT4: glucose transporter type 4; HIF-1α: hypoxia-inducible factor 1 alpha; LPL: lipoprotein lipase; PI3K: phosphatidylinositol 3-kinase; PPARγ: peroxisome proliferator-activated receptor gamma; SREBP-1c: sterol regulatory element-binding protein-1c.

**Figure 4 pharmaceuticals-18-01482-f004:**
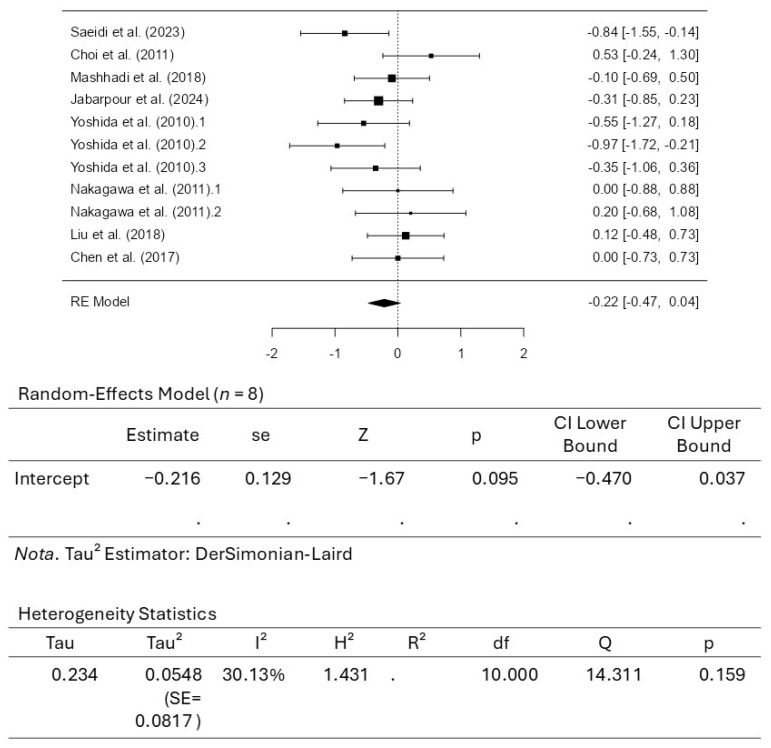
Forrest plot for body mass index (BMI). [41,42,44,45,46,47,48,49].

**Figure 5 pharmaceuticals-18-01482-f005:**
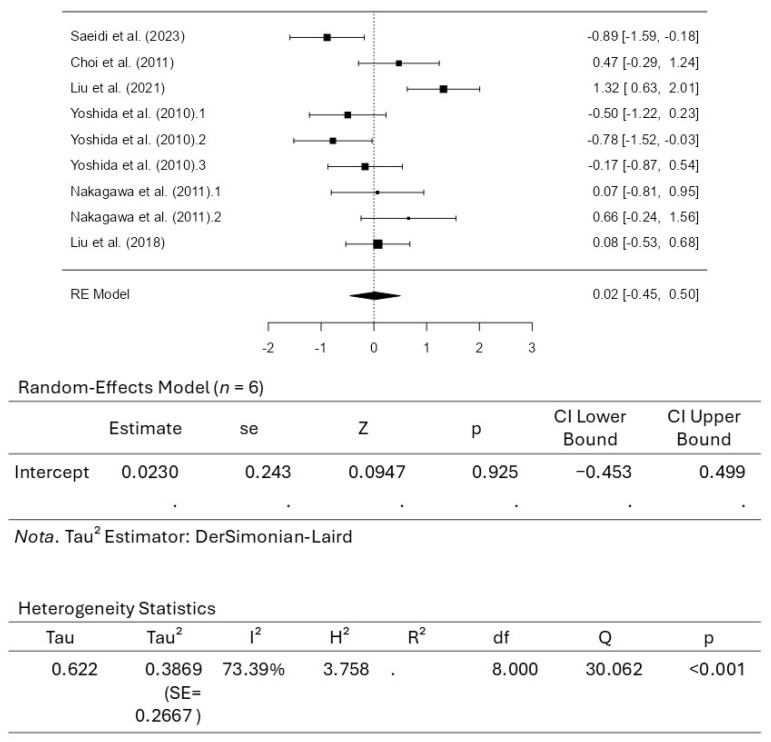
Forrest plot for body weight [42,43,45,46,47,48].

**Figure 6 pharmaceuticals-18-01482-f006:**
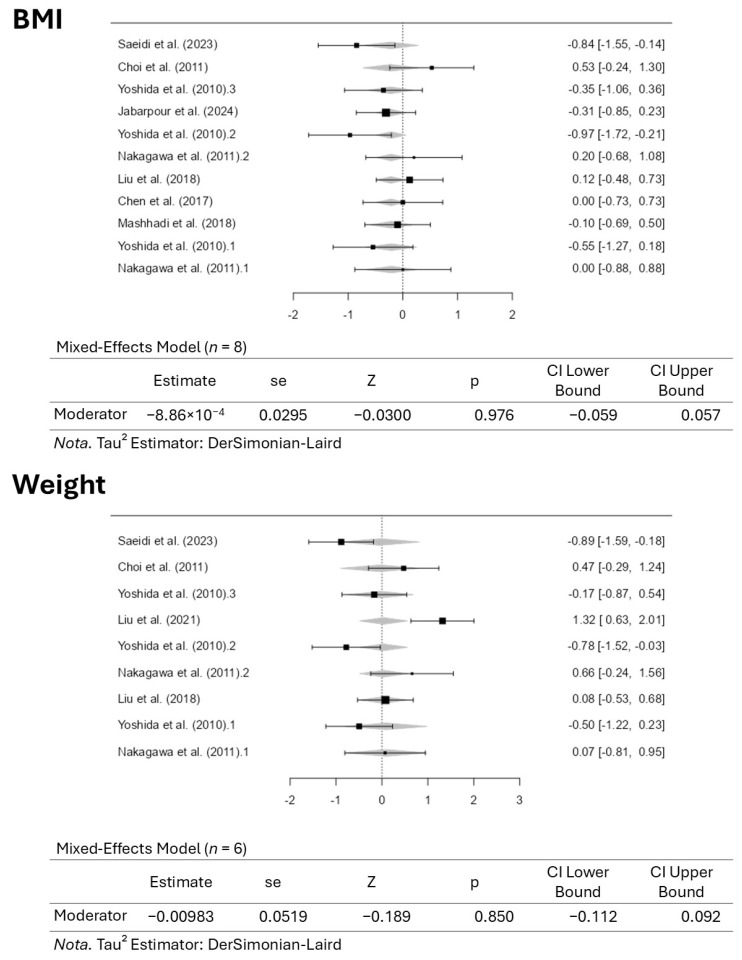
Dose–response meta-analysis for body mass index (BMI) and body weight assessments [41,42,43,44,45,46,47,48,49].

**Figure 7 pharmaceuticals-18-01482-f007:**
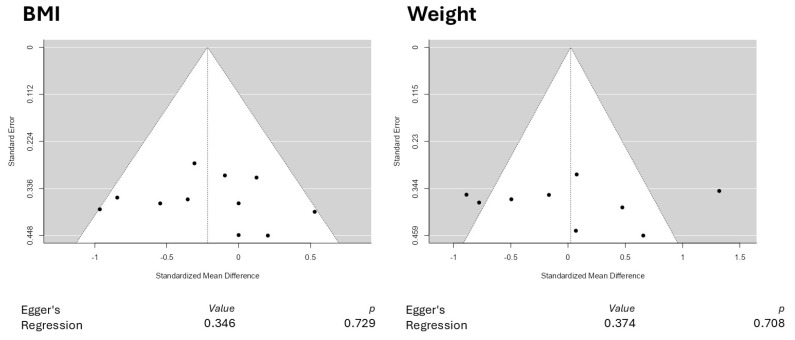
Funnel plots for body mass index (BMI) and body weight assessments.

**Table 1 pharmaceuticals-18-01482-t001:** Report of the included studies following the PRISMA guidelines [36].

Study	Local/Design	Population	Intervention	Comparison	Outcomes	Adverse Effects	Observations
[41]	Iran: randomized, double-blind, placebo-controlled clinical trial.	56 women with PCOS began the study, but 3 were lost to follow-up. Placebo group: 27 participants (age of 31.19 ± 4.57 years; BMI of 26.55 ± 1.89 kg/m^2^). ASX group: 26 participants (age of 30.42 ± 4.69 years; BMI of 26.08 ± 1.89 kg/m^2^).	8 weeks of oral treatment of 12 mg ASX.	Placebo (matching ASX capsules in terms of color, shape, size, packaging, and other attributes, with the same dosage regimen and duration)	ASX had no significant positive effect on BMI. Placebo group: At the study’s end, BMI was 26.51 ± 1.76 kg/m^2^. ASX group: At the study’s end, BMI was 25.89 ± 2.20 kg/m^2^. Mean changes: *p*-value of 0.571.	No adverse effects were observed.	The study has some limitations, including a small sample size, a brief intervention duration, and the lack of quantification of ASX levels in serum or plasma at the trial’s outset.
[42]	Iran: randomized, placebo-controlled study.	68 males with obesity (mean age: 27.6 ± 8.4 years; mean height: 167.8 ± 3.1 cm). Control group: 17 participants, BW of 95.3 ± 1.8 kg, fat percentage of 31.1 ± 1.5%, BMI of 34.1 ± 2.5 kg/m^2^. ASX group: 17 participants, BW of 94.2 ± 2.6 kg, fat percentage of 31.1 ± 1.5%, BMI of 33.2 ± 1.4 kg/m^2^.	20 mg ASX capsule once daily with breakfast for 12 weeks.	Placebo (20 mg/day of raw corn starch, with the same ASX dosage regimen and duration)	BW reductions were significant in the ASX group (*p* = 0.008), but not in the control group (*p* = 0.32). Changes in BMI were significantly decreased in the ASX group (*p* = 0.019), but not in the control group (*p* = 0.37). The decrease in body fat percent was significant in the ASX group (*p* = 0.004), but not in the control group (*p* = 0.28).	Not reported.	Eight participants from different groups withdrew from the trial and it was not specified whether the data from these individuals were considered in the statistical analysis or whether their exclusion could have impacted the results.
[43]	USA: randomized, double-blind, placebo-controlled study.	Healthy older adults. Placebo group: 8 ♂ (age of 74.2 ± 1.6 years; BW of 169.1 ± 12.9 lbs) and 10 ♀ (age of 70.4 ± 1.6 years; BW of 142.1 ± 5.3 lbs). ASX group: 9 ♂ (age of 69.2 ± 1.0 years; BW of 184.2 ± 5.0 lbs) and 13 ♀ (age of 68.7 ± 0.6 years; BW of 145.4 ± 7.4 lbs).	A formulation containing 12 mg of ASX was consumed as two capsules per day, for 12 weeks.	Placebo (with the same ASX duration, but other details were not provided)	Only male participants of the placebo group had a minimal decrease (>1%) in BW. All other groups maintained weight stable (<1% difference). Placebo group: 8 ♂ (at the study’s end, BW was 166.5 ± 12.7 lbs) and 10 ♀ (at the study’s end, BW was 140.7 ± 5.2 lbs). ASX group: 9 ♂ (at the study’s end, BW was 185.8 ± 5.1 lbs) and 13 ♀ (at the study’s end, BW was 144.0 ± 7.1 lbs).	Not reported.	The study did not explore the biochemical mechanisms underlying the effects of ASX on fat and carbohydrate utilization.
[44]	Iran: double-blind, randomized, parallel, placebo-controlled trial.	44 participants with type 2 diabetes mellitus were initially included, but 1 dropped out of the study. Placebo group: 9 ♂ and 13 ♀; age of 54 ± 8 years; BMI of 30.4 ± 5 kg/m^2^; total body fat of 39.7 ± 9.8%. ASX group: 8 ♂ and 14 ♀; age of 51 ± 9.7 years; BMI of 30.0 ± 5.11 kg/m^2^; total body fat of 35.5 ± 10.6%.	A single 8 mg capsule of ASX was administered orally once daily, immediately after lunch, for 8 weeks.	Placebo (8 mg tablet containing inactive ingredients such as dicalcium phosphate, microcrystalline cellulose, stearic acid, silicon dioxide, and magnesium stearate, with the same ASX dosage regimen and duration)	Placebo group: the final body fat was 39.8 ± 8.9% (*p* = 0.05) and the final BMI was 30.4 ± 5.09 kg/m^2^ (*p* = 0.04). ASX group: the final BMI was 29.9 ± 5.18 kg/m^2^ (*p* = 0.61), the final total body fat was 35.8 ± 10.4% (*p* = 0.67).	No adverse effects were observed.	The study was unable to determine the molecular mechanisms by which ASX enhances insulin sensitivity in human cells.
[45]	Korea: randomized, double-blind, placebo-controlled study.	27 overweight adults. Placebo group: 11 ♂ and 2 ♀; age of 30.1 ± 9.5 years; BMI of 26.3 ± 1.3 kg/m^2^; BW of 77.1 ± 10.8 kg, height of 1.71 ± 0.10 m. ASX group: 12 ♂ and 2 ♀; age of 31.1 ± 9.4 years; BMI of 28.1 ± 2.4 kg/m^2^; BW of 83.6 ± 9.4 kg, height of 1.72 ± 0.07 m.	One 20 mg ASX capsule once daily after breakfast for 12 weeks.	Placebo (one capsule daily, but other details were not provided)	BMI and BW values changed minimally in the ASX group, while they both increased in the placebo group. ASX may hinder BW gain, but this study showed no significant difference. Placebo group: at the study’s end, BMI was 27.1 ± 2.2 kg/m^2^ and BW was 79.1 ± 11.5 kg. ASX group: At the study’s end, BMI was 28.3 ± 2.2 kg/m^2^ and BW was 84.1 ± 8.9 kg.	A change in fecal color to red (it could have been due to the reddish color of ASX) was reported by two participants, and two participants reported an increase in bowel movements.	The biological mechanisms underlying the protective effects of ASX, including its antioxidant activity in humans, remain insufficiently understood. Further studies are needed to investigate whether ASX truly has a BW-lowering effect.
[46]	Japan: randomized, double-blind, placebo-controlled study.	61 non-obese subjects with fasting serum triglyceride of 120–200 mg/dL, without diabetes and hypertension. Placebo group: 10 ♂ and 5 ♀; age of 44.3 ± 7.0 years; BMI of 25.1 ± 2.8 kg/m^2^; BW of 69.4 ± 7.9 kg. ASX 6 mg/day group: 10 ♂ and 5 ♀; age of 47.0 ± 7.0 years; BMI of 23.6 ± 3.2 kg/m^2^; BW of 64.6 ± 12.2 kg. ASX 12 mg/day group: 10 ♂ and 5 ♀; age of 42.8 ± 8.8 years; BMI of 23.0 ± 2.2 kg/m^2^; BW of 64.2 ± 7.2 kg. ASX 18 mg/day group: 11 ♂ and 5 ♀; age of 43.8 ± 10.4 years; BMI of 23.9 ± 7.0 kg/m^2^; BW of 67.3 ± 5.6 kg.	12-week treatment of 6, 12, and 18 mg/day of ASX.	Placebo (details about it were not provided)	BMI and BW were unaffected at all doses. ASX 6 mg/day group: at the study’s end, BMI was 23.7 ± 3.0 kg/m^2^; BW was 64.8 ± 11.7 kg. ASX 12 mg/day group: at the study’s end, BMI was 22.9 ± 2.1 kg/m^2^; BW was 63.9 ± 7.0 kg. ASX 18 mg/day group: at the study’s end, BMI was 24.2 ± 3.3 kg/m^2^; BW was 68.1 ± 12.4 kg.	Not reported.	The study did not reveal the specific underlying mechanisms by which ASX improves lipid metabolism and prevents atherosclerosis. Moreover, well-defined in vitro studies and long-term and large-scale clinical trials are essential to confirm ASX’s effects on human health.
[47]	Japan: randomized, double-blind, placebo-controlled study.	30 healthy adults. Placebo group: 5 ♂ and 5 ♀; age of 56.6 ± 4.4 years; BMI of 27.7 ± 2.1 kg/m^2^; BW of 70.3 ± 9.3 kg; height of 159 ± 11 cm. ASX 6 mg/day group: 5 ♂ and 5 ♀; age of 56.3 ± 6.6 years; BMI of 27.4 ± 2.2 kg/m^2^; BW of 70.5 ± 8.1 kg; height of 160 ± 8 cm. ASX 12 mg/day group: 5 ♂ and 5 ♀; age of 56.1 ± 5.1 years; BMI of 27.6 ± 2.1 kg/m^2^; BW of 74.4 ± 5.3 kg; height of 164 ± 7 cm.	12-week treatment of 6 or 12 mg/day of ASX.	Placebo (capsules containing maize oil and colored to appear identical to ASX capsules)	The study did not report statistical analysis for the changes in BW and BMI. Placebo group: at the study’s end, BMI was 27.1 ± 2.2 kg/m^2^ and BW was 69.0 ± 9.2 kg. ASX 6 mg/day group: At the study’s end, BMI was 27.1 ± 2.2 kg/m^2^ and BW was 69.6 ± 7.5 kg. ASX 12 mg/day group: At the study’s end, BMI was 27.6 ± 2.5 kg/m^2^ and BW was 74.3 ± 5.9 kg.	No serious side effects were reported.	Further studies are needed to thoroughly investigate the action of ASX and its antioxidant role and potential anti-metabolic syndrome effects.
[48]	USA: randomized, double-blind, placebo-controlled trial.	58 healthy elderly individuals began the study, but 16 dropped out. Placebo group: 10 ♂ and 9 ♀; age of 72.2 ± 5.2 years; BMI of 24.7 ± 3.1 kg/m^2^; BW of 71.1 ± 14.8 kg. ASX group: 10 ♂ and 13 ♀; age of 69.1 ± 3.4 years; BMI of 26.3 ± 3.2 kg/m^2^; BW of 73.8 ± 13.4 kg.	A dietary formulation containing 12 mg of ASX was consumed as two capsules per day. The intervention lasted 4 months.	Placebo (details about it were not provided)	The study did not report statistical analysis for the changes in BW and BMI. Placebo group: at the study’s end, BMI was 25.4 ± 3.1 kg/m^2^ and BW was 71.3 ± 14.8 kg. ASX group: At the study’s end, BMI was 25.8 ± 3.2 kg/m^2^, and BW was 72.4 ± 13.9 kg.	No serious side effects were observed.	The study does not describe the inflammatory and oxidative stress biomarkers by which ASX ameliorates the age-associated decline in muscle strength and size.
[49]	Japan: randomized, double-blind, placebo-controlled study.	29 healthy women in a climacteric phase. Placebo group: 15 subjects; age of 52 ± 4 years; BMI of 21.9 ± 4.6 kg/m^2^. ASX group: 14 subjects; age of 51 ± 5 years; BMI of 21.3 ± 2.7 kg/m^2^.	12 mg/day of ASX for 3 months.	Placebo (details about it were not provided).	The changes in BMI were not statistically significant (*p* = 0.81). Placebo group: at the study’s end, BMI was 21.5 ± 4.6 kg/m^2^. ASX group: At the study’s end, BMI was 21.5 ± 2.7 kg/m^2^.	Not reported.	The study suggests that ASX may exert hepatoprotective effects, but its specific mechanisms in mitigating obesity-related liver dysfunction remain unclear.

Abbreviations: ASX, Astaxanthin; BMI, Body Mass Index; BW, Body Weight; PCOS, Polycystic Ovary Syndrome.

**Table 2 pharmaceuticals-18-01482-t002:** Report of bias identification throughout the studies following the COCHRANE Handbook for Intervention Assessment [37].

Study	Question Focus	Appropriate Randomization	Allocation Blinding	Double-Blind	Losses (<20%)	Prognostic or Demographic Characteristics	Outcomes	Intention to Treat Analysis	Sample Calculation	Adequate Follow-Up
[41]		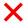						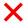		
[42]		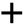	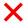	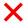		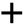		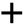		
[43]					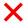			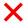	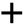	
[44]		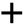						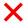	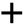	
[45]		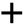								
[46]		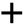							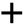	
[47]		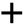							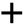	
[48]					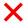			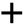	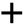	
[49]		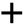							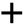	

Abbreviations: 

, Low risk of bias; 
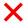
, High risk of bias; 
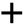
, Unclear risk of bias.

**Table 3 pharmaceuticals-18-01482-t003:** Summary of key study differences for stratified analysis.

Stratification Category	Observed Variations Across the Included Studies
Astaxanthin dosage	Typically ranged from 6 mg/day to 20 mg/day.
Intervention duration	Ranged from 8 weeks to 4 months (12 weeks in most studies; one study lasted 4 months).
Population age	Ranged from young adults to elderly individuals.
Health status	Included healthy individuals, individuals with ovarian diseases, obesity, type 2 diabetes, and those with elevated triglycerides.
Intervention protocols	Varied principally in frequency and administration timing.
Subgroup analysis performed	Conducted mainly based on astaxanthin dosage or treatment vs. placebo and health status.

## Data Availability

The raw data supporting the conditions of this meta-analysis will be made available for readers upon reasonable request to the corresponding author.

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
