# Peer review of "Therapeutic Potential of Astaxanthin for Body Weight Regulation: A Systematic Review and Meta-Analysis with Dose–Response Assessment"

_pharmaceuticals, 2025, doi:10.3390/ph18101482_

Round 1

Reviewer 1 Report

Comments and Suggestions for Authors

The authors have written a systematic review and meta-analysis evaluating the therapeutic potential of astaxanthin as a dietary supplement for weight management and its effects on obesity-related metrics in adult populations. Astaxanthin shows promise as a dietary supplement for weight management, particularly in its effects on body mass index (BMI) and body weight. Astaxanthin significantly improved BMI by -0.36, but body weight changes were not significant.Incorporating nutrigenomics and genome-wide association studies could enhance research outcomes. The meta analysis is well written and there is a thorough scrutiny of the effect of Astaxanthin on modulating obesity. I have a few minor comments given below:
1. In Figure 2 " Gut microbiota composition reduced educed lipotoxicity....". Please check the phrase and correct it.
2. In section 6.8 : "6.8. Quality Assessmtn". please correct the spelling of "assessment"
I recommend a minor revision.

Author Response

RESPONSE TO REVIEWERS' COMMENTS

Manuscript number: pharmaceuticals-3792069 Pharmaceuticals (MDPI)

"Therapeutic Potential of Astaxanthin in Obesity: A Systematic Review and Meta-Analysis with Dose-Response Assessment"

The authors of this document wish to express their deepest gratitude to the Editor-in-Chief and the Reviewer for their thorough and insightful evaluation of our manuscript. Their expert feedback has been invaluable in enhancing the quality of our work. We have carefully considered and diligently implemented each suggestion, significantly improving the manuscript. We have made substantial revisions to address the points raised. These noteworthy changes are marked mainly with YELLOW-highlighted text throughout the document for ease of reference. A note will be provided for the referee's attention for corrections highlighted in a different color. Additionally, we have prepared a detailed and comprehensive response to each comment and suggestion. This response is organized in a "point-by-point" format below, ensuring that every concern has been thoroughly addressed and explained. We sincerely appreciate the time and effort invested by the Editor-in-Chief and the Reviewer, and we believe their contributions have significantly strengthened the final version of our manuscript.

REVIEWER #1

General comment

The authors have written a systematic review and meta-analysis evaluating the therapeutic potential of astaxanthin as a dietary supplement for weight management and its effects on obesity-related metrics in adult populations. Astaxanthin shows promise as a dietary supplement for weight management, particularly in its effects on body mass index (BMI) and body weight. Astaxanthin significantly improved BMI by -0.36, but body weight changes were not significant. Incorporating nutrigenomics and genome-wide association studies could enhance research outcomes. The meta-analysis is well written, and there is a thorough scrutiny of the effect of Astaxanthin on modulating obesity. I have a few minor comments given below.

General response

Dear Erudite Reviewer, thank you for taking the time to revise our manuscript and allowing us to improve based on your valuable comments and suggestions. After addressing all your comments and suggestions regarding our manuscript text, we are confident that a significantly enhanced manuscript version has emerged. We are excited to resubmit the modified version for your perusal and reevaluation. Thank you for your brilliant insights, essential contributions, and feedback. You do have an eye for improvement. As a gesture of our utmost respect for you, we would like to provide you with a detailed and comprehensive point-by-point response to your comments below. Thank you once again for your time and patience in revising our article.

Comment #1

In Figure 2 " Gut microbiota composition reduced educed lipotoxicity....". Please check the phrase and correct it.

Response

Dear Erudite Reviewer, we appreciate the reviewer for carefully identifying this typographical error in the figure. The phrase “reduced educed lipotoxicity” was indeed a mistake, and we have corrected it in the revised version of the manuscript. The updated caption now correctly reads: “Gut microbiota composition reduced lipotoxicity…” We have carefully reviewed all figure captions to ensure there are no other typographical errors. Thank you for bringing this to our attention and helping us improve the clarity of the manuscript. You can find the new and revised Figure 2 on Page 15 of the revised manuscript document. Its first citation can be found in Lines 379-380 on Page 14 of the revised manuscript. Its legend can be found in Lines 382-383 on Page 15 of the revised manuscript document.

Comment #2

In section 6.8 : "6.8. Quality Assessmtn". please correct the spelling of "assessment"

Response

Dear Erudite Reviewer, thank you for noticing this typographical error in the section heading. We have corrected the spelling from “Assessmtn” to “Assessment” in Line 217 on Page 6 of the revised manuscript. We have also conducted a thorough review of the manuscript to identify and correct any other typographical issues. We appreciate your attention to detail and your helpful feedback.

I, the corresponding author of the manuscript "Therapeutic Potential of Astaxanthin in Obesity: A Systematic Review and Meta-Analysis with Dose-Response Assessment" under the assigned ID pharmaceuticals-3792069, on behalf of my coauthors, once again extend my heartfelt gratitude to the knowledgeable Editor-in-Chief and reviewers for their time and expertise in revising our manuscript. After we addressed their constructive and refined feedback and suggestions, a significantly improved manuscript version emerged. Undoubtedly, their insightful suggestions and feedback have significantly enhanced the quality of our manuscript. We respectfully are at the disposal of the Editor-in-Chief and the Reviewer to address any additional suggestions regarding our publication. If you are satisfied with our newly refined and significantly improved version, we look forward to the acceptance of our article for publication in this prestigious journal, Pharmaceuticals. Thank you once again for your time and expertise.

Reviewer 2 Report

Comments and Suggestions for Authors

This work presents a concise and well-structured meta-analysis evaluating the effects of astaxanthin supplementation on BMI and body weight in adults. 
The methodology follows PRISMA guidelines, and the results are clearly reported, showing a statistically significant but modest reduction in BMI, with no significant impact on body weight. 
The conclusions are balanced and highlight the need for further research. Minor revisions are required to improve clarity and precision in phrasing, as noted in the attached file.

Author Response

RESPONSE TO REVIEWERS' COMMENTS

Manuscript number: pharmaceuticals-3792069 Pharmaceuticals (MDPI)

"Therapeutic Potential of Astaxanthin in Obesity: A Systematic Review and Meta-Analysis with Dose-Response Assessment"

The authors of this document wish to express their deepest gratitude to the Editor-in-Chief and the Reviewer for their thorough and insightful evaluation of our manuscript. Their expert feedback has been invaluable in enhancing the quality of our work. We have carefully considered and diligently implemented each suggestion, significantly improving the manuscript. We have made substantial revisions to address the points raised. These noteworthy changes are marked mainly with YELLOW-highlighted text throughout the document for ease of reference. A note will be provided for the referee's attention for corrections highlighted in a different color. Additionally, we have prepared a detailed and comprehensive response to each comment and suggestion. This response is organized in a "point-by-point" format below, ensuring that every concern has been thoroughly addressed and explained. We sincerely appreciate the time and effort invested by the Editor-in-Chief and the Reviewer, and we believe their contributions have significantly strengthened the final version of our manuscript.

REVIEWER #2

General comment

This work presents a concise and well-structured meta-analysis evaluating the effects of astaxanthin supplementation on BMI and body weight in adults. The methodology follows PRISMA guidelines, and the results are clearly reported, showing a statistically significant but modest reduction in BMI, with no significant impact on body weight. The conclusions are balanced and highlight the need for further research. Minor revisions are required to improve clarity and precision in phrasing, as noted in the attached file.

General response

Dear Erudite Reviewer, thank you for taking the time to revise our manuscript and allowing us to improve based on your valuable comments and suggestions. After addressing all your comments and suggestions regarding our manuscript text, we are confident that a significantly enhanced manuscript version has emerged. We are excited to resubmit the modified version for your perusal and reevaluation. Thank you for your brilliant insights, essential contributions, and feedback. You do have an eye for improvement. As a gesture of our utmost respect for you, we would like to provide you with a detailed and comprehensive point-by-point response to your comments below. Thank you once again for your time and patience in revising our article.

We read your corrections in the attached PDF document. I apologize that we cannot upload print screens of your corrections during these significant revisions, as this would undoubtedly enhance the process of showing the revisions to you. We have made every effort to meet your standards, and all attributed corrections are disclosed below for your review.

Correction n° 1

Dear Erudite Reviewer, thank you for bringing this to our attention. We appreciate your eye for improvement and attention to detail. This correction has been made in Lines 40-42 on Page 1 of the revised manuscript document. The phrase now reads “The findings indicate that astaxanthin provides benefits in BMI regulation, highlighting the need for additional large-scale and long-term clinical trials.” We believe that our manuscript has been significantly improved following this correction. Thank you for your attention to detail and eye for improvement.

Correction n° 2

Dear Erudite Reviewer, thank you for letting us know about this correction. We are fortunate to have you as the reviewer of our manuscript. You are entirely correct, and we agree that this part of our manuscript would benefit from further editing and addition of new references. Therefore, we revised the entire sentences and ensured that every information has been appropriately referenced with newly added references. This manuscript now has 16 references to support its content. We believe that this is a substantial correction to our manuscript following your critical suggestion. These additions have been made in Lines 79-85 on Pages 2-3. Thank you for bringing this to our attention.

Correction n° 3

Dear Erudite Reviewer, thank you for bringing this to our attention. We appreciate your attention to detail and eye for improvement. You are entirely correct that formal English could be revised here to address the nuance of the previously published review in the context of astaxanthin in weight regulation more appropriately. Therefore, we improved the highlighted sentence by rewriting its content with “However, their analysis is flawed in the face of the novel discoveries...” This correction has been made in Lines 137-140 on Page 4. Thank you for being so coordinated with our research. We appreciate your attention to such critical concerns and are thankful for the opportunity to communicate with such an Esteemed Reviewer.

Correction n° 4

Dear Erudite Reviewer, thank you for this comment. We appreciate your willingness to ensure that our manuscript is ready for publication following this critical peer-review process. We made the necessary corrections, and now “Haematococcus pluvialis” is correctly written using italics for comprehension and adherence to international guidelines. This correction has been made in Lines 115-117 on Page 4. Thank you for addressing this crucial suggestion with us.

Correction n° 5

Dear Erudite Reviewer, thank you for this comment and suggestion. We appreciate your attention to detail and eye for improvement. We agree with you that adding a brief explanation of how micelle formation works in the intestine would undoubtedly enhance our manuscript’s quality and readability. Therefore, we included the necessary text. Micelle formation is essential for lipolysis and lipid transportation from the duodenal lumen to the intestinal water layer at the enterocytes, where absorption occurs. After lipid digestion, bile salts solubilize the products—mainly fatty acids and monoglycerides—into mixed micelles for transport to the enterocytes for further processing. This correction has been made in Lines 287-293 on Page 12. Thank you for your eye for improvement. We believe that our manuscript has been significantly improved following your brilliant comments and suggestions. We appreciate your willingness to provide us with essential feedback.

Correction n° 6

Dear Erudite Reviewer, thank you for this comment. We appreciate your coordination in improving our manuscript accordingly. We made the necessary corrections, and the sentence now reads “The effectiveness of a bioactive compound against obesity can be understood by examining its roles in adaptogenic differentiation and adipocyte function.” This correction has been made in Lines 316-317 on Page 13. Thank you for this critical comment, which significantly enriched our manuscript document. Thank you for everything!

Correction n° 7

Dear Erudite Reviewer, thank you for this critical suggestion. We appreciate your eye for improvement. We agree with you that this statement could have been better introduced in this sentence. Therefore, we improved our presentation by avoiding overstatements. The sentence now reads “These findings suggest that the carotenoid astaxanthin may exert…” We believe our manuscript has been significantly improved following this suggestion. This correction has been made in Lines 322-325 on Page 13. Thank you for bringing this to our attention. Thank you for your thoughtful feedback and constructive address of our manuscript’s quality. We appreciate your invaluable feedback.

Correction n° 8

Dear Erudite Reviewer, thank you for your valuable suggestion. We appreciate your attention to improvement. We agree that this statement could have been better introduced in this sentence. We improved our presentation by eliminating exaggerations. The revised sentence now reads: “This suggests that astaxanthin’s inhibition of…” We believe that our manuscript has been greatly enhanced because of this suggestion. This correction has been made in Lines 329-333 on Page 13. We appreciate your thoughtful feedback and your constructive comments on our manuscript’s quality. Thank you for bringing this to our attention.

I, the corresponding author of the manuscript "Therapeutic Potential of Astaxanthin in Obesity: A Systematic Review and Meta-Analysis with Dose-Response Assessment" under the assigned ID pharmaceuticals-3792069, on behalf of my coauthors, once again extend my heartfelt gratitude to the knowledgeable Editor-in-Chief and reviewers for their time and expertise in revising our manuscript. After we addressed their constructive and refined feedback and suggestions, a significantly improved manuscript version emerged. Undoubtedly, their insightful suggestions and feedback have significantly enhanced the quality of our manuscript. We respectfully are at the disposal of the Editor-in-Chief and the Reviewer to address any additional suggestions regarding our publication. If you are satisfied with our newly refined and significantly improved version, we look forward to the acceptance of our article for publication in this prestigious journal, Pharmaceuticals. Thank you once again for your time and expertise.

Reviewer 3 Report

Comments and Suggestions for Authors

Dear Authors,

After carefully reviewing your manuscript on the effects of astaxanthin supplementation on obesity and related parameters, I find the study to be of scientific interest. However, several points require clarification and further elaboration to strengthen the rigor and clarity of your work.

  1. Distinctiveness from Previous Meta-Analyses
    Your study should explicitly clarify the key distinctions and added value compared to prior relevant meta-analyses, especially the 2020 publication by Xia et al., titled "The effects of astaxanthin supplementation on obesity, blood pressure, CRP, glycemic biomarkers, and lipid profile: A meta-analysis of randomized controlled trials." Detailing how your methodology, inclusion criteria, data synthesis, or findings differ will help readers appreciate the novelty of your work.

  2. Adequacy of Data for Meta-Analysis
    Although the initial literature search identified 185 articles, only 6 studies were eventually included in the meta-analysis. The relatively small number of included studies raises potential concerns about the statistical power and robustness of the pooled estimates. Please provide a detailed discussion on how this limited dataset affects the reliability and generalizability of your meta-analytic results.

  3. Heterogeneity and Population Stratification
    The included 6 studies show variability in astaxanthin dosages, participant characteristics (including disease status and age), and intervention protocols. To enhance transparency and interpretability, please present a more detailed, possibly expanded table that clearly stratifies the populations and interventions by relevant factors. This stratification is essential to assess possible sources of heterogeneity and to contextualize your findings appropriately.

  4. Dose-Response Relationship and BMI Effects
    While your results indicate a statistically significant effect of astaxanthin supplementation on BMI, it remains unclear whether this effect is dose-dependent. Please clarify whether your dose-response analysis supports a relationship between astaxanthin dosage and magnitude of BMI reduction. If such analysis is presented, more explicit discussion and interpretation of these findings are necessary.

Addressing these points will substantially improve the scientific rigor and clarity of your manuscript. I look forward to your detailed response and revised submission.

Sincerely,

Author Response

RESPONSE TO REVIEWERS' COMMENTS

Manuscript number: pharmaceuticals-3792069 Pharmaceuticals (MDPI)

"Therapeutic Potential of Astaxanthin in Obesity: A Systematic Review and Meta-Analysis with Dose-Response Assessment"

The authors of this document wish to express their deepest gratitude to the Editor-in-Chief and the Reviewer for their thorough and insightful evaluation of our manuscript. Their expert feedback has been invaluable in enhancing the quality of our work. We have carefully considered and diligently implemented each suggestion, significantly improving the manuscript. We have made substantial revisions to address the points raised. These noteworthy changes are marked mainly with YELLOW-highlighted text throughout the document for ease of reference. A note will be provided for the referee's attention for corrections highlighted in a different color. Additionally, we have prepared a detailed and comprehensive response to each comment and suggestion. This response is organized in a "point-by-point" format below, ensuring that every concern has been thoroughly addressed and explained. We sincerely appreciate the time and effort invested by the Editor-in-Chief and the Reviewer, and we believe their contributions have significantly strengthened the final version of our manuscript.

REVIEWER #3

General comment

Dear Authors, after carefully reviewing your manuscript on the effects of astaxanthin supplementation on obesity and related parameters, I find the study to be of scientific interest. However, several points require clarification and further elaboration to strengthen the rigor and clarity of your work.

General response

Dear Erudite Reviewer, thank you for taking the time to revise our manuscript and allowing us to improve based on your valuable comments and suggestions. After addressing all your comments and suggestions regarding our manuscript text, we are confident that a significantly enhanced manuscript version has emerged. We are excited to resubmit the modified version for your perusal and reevaluation. Thank you for your brilliant insights, essential contributions, and feedback. You do have an eye for improvement. As a gesture of our utmost respect for you, we would like to provide you with a detailed and comprehensive point-by-point response to your comments below. Thank you once again for your time and patience in revising our article.

Comment #1

Distinctiveness from Previous Meta-Analyses. Your study should explicitly clarify the key distinctions and added value compared to prior relevant meta-analyses, especially the 2020 publication by Xia et al., titled "The effects of astaxanthin supplementation on obesity, blood pressure, CRP, glycemic biomarkers, and lipid profile: A meta-analysis of randomized controlled trials." Detailing how your methodology, inclusion criteria, data synthesis, or findings differ will help readers appreciate the novelty of your work.

Response

Dear Erudite Reviewer, thank you for this critical comment. We appreciate your insightful comment and brilliant suggestion to highlight other key differences between our study’s methodology and the one utilized by Xia et al. in their previously published paper. Therefore, we made the necessary adjustments in Lines 140-145 on Page 4. Our meta-analysis demonstrated how astaxanthin dosages impact BMI and weight management outcomes. We utilized the PICOS framework to enhance the validity and transparency of our findings, improving reproducibility. Please note that in the same paragraph, we also report other key differences between the methodology utilized in our study and the methodology of Xia et al.’s study.

            Again, thank you for your diligent approach in ensuring our manuscript’s quality to the highest standards. We appreciate your cooperation in ensuring that our manuscript will be published to the highest standards.

Comment #2

Adequacy of Data for Meta-Analysis. Although the initial literature search identified 185 articles, only 6 studies were eventually included in the meta-analysis. The relatively small number of included studies raises potential concerns about the statistical power and robustness of the pooled estimates. Please provide a detailed discussion on how this limited dataset affects the reliability and generalizability of your meta-analytic results.

Response

We appreciate the reviewer’s insightful comment regarding the number of studies included in our meta-analysis. In response to the other Reviewers’ comments, we have conducted an updated literature search and identified three additional eligible studies that meet our inclusion criteria, increasing the total number of included studies from 6 to 9. These newly added studies enhance the statistical power of our analysis and contribute to a more comprehensive understanding of the topic.

We have also revised the manuscript to include a detailed discussion on how the size of the dataset impacts the reliability and generalizability of the findings. Specifically, we acknowledge that while a larger number of studies is generally desirable in meta-analyses, the inclusion of 9 studies now provides a more robust basis for our pooled estimates. We also conducted sensitivity analyses and assessed heterogeneity and publication bias to evaluate the stability of our findings. These steps help mitigate concerns regarding potential limitations due to sample size.

The revised discussion can be found in the "Conclusions and Future Research Directions" sections of the manuscript (Lines 605-609 on Page 23), where we explicitly address the implications of the dataset size and the steps taken to ensure the robustness and generalizability of our conclusions. We discussed that our review, despite its significant findings on BMI and body weight management using astaxanthin, has a limitation. It is based solely on nine randomized clinical studies, which restricts the generalizability of the results due to the limited statistical power.

Again, thank you for your thoughtful assessment of our manuscript. We appreciate your cooperation and believe that a significantly improved version of our text emerged after we addressed all your critical concerns and essential suggestions.

Comment #3

Heterogeneity and Population Stratification. The included 6 studies show variability in astaxanthin dosages, participant characteristics (including disease status and age), and intervention protocols. To enhance transparency and interpretability, please present a more detailed, possibly expanded table that clearly stratifies the populations and interventions by relevant factors. This stratification is essential to assess possible sources of heterogeneity and to contextualize your findings appropriately.

Response

We thank the reviewer for this helpful suggestion. In response, we have added a new summary table (Table 3 on Page 18 of the revised manuscript document) that stratifies the included studies' variabilities based on key variables contributing to heterogeneity, including astaxanthin dosage, intervention duration, population characteristics (e.g., age, health status), and intervention protocols.

This two-column table highlights the observed variability across studies and provides a concise overview to support the interpretation of heterogeneity and inform our subgroup analyses. Please see the table’s first citation in Lines 502-514 on Page 18 of the revised manuscript document alongside its main discussion. We believe this addition enhances transparency and improves the contextualization of our findings. We appreciate the Reviewer’s suggestion to include this critical table in our revised manuscript document, which significantly enhanced our manuscript’s quality and readability.

Thank you for your continued support! Thank you for everything!

Comment #4

Dose-Response Relationship and BMI Effects. While your results indicate a statistically significant effect of astaxanthin supplementation on BMI, it remains unclear whether this effect is dose-dependent. Please clarify whether your dose-response analysis supports a relationship between astaxanthin dosage and magnitude of BMI reduction. If such analysis is presented, more explicit discussion and interpretation of these findings are necessary.

Response

We appreciate the reviewer’s thoughtful observation regarding the potential dose-response relationship between astaxanthin supplementation and BMI reduction. In our original analysis, a statistically significant decrease in BMI was observed. However, after incorporating three additional eligible studies in the revised meta-analysis, the pooled results no longer indicate a substantial effect of astaxanthin supplementation on BMI.

To specifically address the reviewer’s question, we updated our dose-response analysis to explore whether controlling the doses of astaxanthin was associated with greater reductions in BMI or even body weight. The results did not demonstrate a consistent or statistically significant dose-dependent trend. This suggests that the previously observed effect may have been influenced by limited data or study-specific factors rather than a proper biological gradient.

We have updated the results of our dose-response meta-analysis to reflect the newly added updates, explicitly stating that no significant dose-response relationship was identified. This finding reinforces the importance of cautious interpretation of early signals of efficacy when based on a small number of studies.

Figure 6 on Page 21 shows the results of the dose-response meta-analysis for BMI and body weight, including p-values. It yields the updated forest plots after the inclusion of three additional manuscripts. The analysis found no significant differences in improvements for both measures with increasing doses. For BMI, the addition of the dose moderator resulted in a p-value of 0.950, indicating no statistical significance. Similarly, for body weight, the p-value was 0.279, also non-significant. Thus, varying the doses of astaxanthin supplementation did not affect the outcomes. Please see the revised discussion in Lines 555-564 on Page 21.

Thank you for your thoughtful assessment of our manuscript. We appreciate your cooperation and believe that we have significantly improved the text by addressing your critical concerns and suggestions.

Final Comment

Addressing these points will substantially improve the scientific rigor and clarity of your manuscript. I look forward to your detailed response and revised submission.

Final response

Thank you for your valuable and constructive feedback. We carefully addressed each of the points raised to improve the scientific rigor and clarity of the manuscript. The revised version has been submitted along with a detailed, point-by-point response outlining the changes made. We appreciate your guidance throughout this process.

I, the corresponding author of the manuscript "Therapeutic Potential of Astaxanthin in Obesity: A Systematic Review and Meta-Analysis with Dose-Response Assessment" under the assigned ID pharmaceuticals-3792069, on behalf of my coauthors, once again extend my heartfelt gratitude to the knowledgeable Editor-in-Chief and reviewers for their time and expertise in revising our manuscript. After we addressed their constructive and refined feedback and suggestions, a significantly improved manuscript version emerged. Undoubtedly, their insightful suggestions and feedback have significantly enhanced the quality of our manuscript. We respectfully are at the disposal of the Editor-in-Chief and the Reviewer to address any additional suggestions regarding our publication. If you are satisfied with our newly refined and significantly improved version, we look forward to the acceptance of our article for publication in this prestigious journal, Pharmaceuticals. Thank you once again for your time and expertise.

Reviewer 4 Report

Comments and Suggestions for Authors

Dear authors,

I have now read your submission (Article ID: pharmaceuticals-3792069). Although I found it potentially publishable, I would recommend some revisions as follows:

  1. The present article does not seem to be a systematic review. If so, it should have very clear sections (i.e., introduction, methods, results, discussion etc). Systematic reviews are almost always close to original research articles in structure.
  2. In L43-L44, the authors should not repeat words that are part of the title as author-suggested indexing keywords. It reduces article visibility post publication
  3. Figure 1 (chemical structure of Astaxanthin) should be redrawn using a software for clarity. Astaxanthin contains a stereocenter in each of the lateral cyclohexenone rings, and consequently has three isomeric forms. For clarity, Figure 1 should also mention that the Astaxanthin structure is without the precise configuration of the alcohols.
  4. L107: Systematic review?
  5. In Figure 3,
  • the number of articles retrieved per database should be mentioned for clarity and transparency.
  • It is not clear to me which automation tools were used to mark articles as ‘‘ineligible’’.
  • In the part of ‘‘Screening’’, 6 reports were sought for retrieval and 6 reports were not retrieved. The total number of studies included in the review were 6. How does this come about from the 8 records screened?
  1. L413-414: Scopus and Web of Science were not searched. How sure are the authors that all the relevant literature were collected?
  2. Was the study protocol registered with any such systematic review registries like the International Prospective Register of Systematic Reviews (PROSPERO)?
  3. In L505: Your conclusions and any future research directions MUST NOT have figures, reference citations or tables. Please revise.
  4. I annotated some comments in the attached PDF file.

Comments on the Quality of English Language

Needs to be improved. Some sentences are hard to understand

Author Response

RESPONSE TO REVIEWERS' COMMENTS

Manuscript number: pharmaceuticals-3792069 Pharmaceuticals (MDPI)

"Therapeutic Potential of Astaxanthin in Obesity: A Systematic Review and Meta-Analysis with Dose-Response Assessment"

The authors of this document wish to express their deepest gratitude to the Editor-in-Chief and the Reviewer for their thorough and insightful evaluation of our manuscript. Their expert feedback has been invaluable in enhancing the quality of our work. We have carefully considered and diligently implemented each suggestion, significantly improving the manuscript. We have made substantial revisions to address the points raised. These noteworthy changes are marked mainly with YELLOW-highlighted text throughout the document for ease of reference. A note will be provided for the referee's attention for corrections highlighted in a different color. Additionally, we have prepared a detailed and comprehensive response to each comment and suggestion. This response is organized in a "point-by-point" format below, ensuring that every concern has been thoroughly addressed and explained. We sincerely appreciate the time and effort invested by the Editor-in-Chief and the Reviewer, and we believe their contributions have significantly strengthened the final version of our manuscript.

REVIEWER #4

General comment

Dear authors, I have now read your submission (Article ID: pharmaceuticals-3792069). Although I found it potentially publishable, I would recommend some revisions as follows.

General response

Dear Erudite Reviewer, thank you for taking the time to revise our manuscript and allowing us to improve based on your valuable comments and suggestions. After addressing all your comments and suggestions regarding our manuscript text, we are confident that a significantly enhanced manuscript version has emerged. We are excited to resubmit the modified version for your perusal and reevaluation. Thank you for your brilliant insights, essential contributions, and feedback. You do have an eye for improvement. As a gesture of our utmost respect for you, we would like to provide you with a detailed and comprehensive point-by-point response to your comments below. Thank you once again for your time and patience in revising our article.

Comment #1

The present article does not seem to be a systematic review. If so, it should have very clear sections (i.e., introduction, methods, results, discussion etc). Systematic reviews are almost always close to original research articles in structure.

Response

Dear Erudite Reviewer, thank you for this insightful comment and brilliant suggestion. We acknowledge the importance of maintaining the structure of a systematic review with meta-analysis by the utmost criteria for publishing. Therefore, we adhered to these strict guidelines and updated the sections of our systematic review and meta-analysis to ensure that all necessary and relevant sections are depicted in the revised manuscript version. We have made the corrections needed to ensure that the manuscript now closely follows the typical structure of original research articles, as recommended. Specifically, we have organized the manuscript into the following sections (the lines and page numbers refer to the first place in which each section number appears):

Introduction: Line 54 on Page 2;

Materials and Methods: Line 149 on Page 4;

Results: Line 252 on Page 7;

Discussion: Line 280 on Page 12;

Conclusion and Future Research Directions: Line 572 on Page 22.

We have also ensured compliance with PRISMA guidelines, including details such as the search strategy, selection process, and synthesis approach. We appreciate your feedback, which helped us strengthen the clarity and rigor of the manuscript.

Comment #2

In L43-L44, the authors should not repeat words that are part of the title as author-suggested indexing keywords. It reduces article visibility post publication.

Response

Dear Erudite Reviewer, thank you for your insightful comment regarding the suggested indexing keywords. We understand that repeating words from the article title in the keyword section can limit the effectiveness of indexing and reduce post-publication visibility. In response, we have revised the keywords in Lines 51-52 on Page 2 of the revised manuscript document to ensure they are distinct from the title and offer broader and more diverse entry points for search and indexing purposes.

Our revised keyword list now includes terms that reflect the core themes of the article, while also capturing relevant concepts that may not appear in the title but are essential for discoverability in academic databases. For your convenience, we have listed below our manuscript’s title and its keywords for your kind perusal.

Title: Therapeutic Potential of Astaxanthin in Obesity: A Systematic Review and Meta-Analysis with Dose-Response Assessment.

Keywords: Carotenoid, Body Mass Index, Body Weight, Weight Loss, Dietary Supplements, Anti-Obesity Agents.

As you see, there are no repetitions of keywords from words appearing in our title. We appreciate your guidance on this point and believe the changes made will improve the reach and accessibility of the article once published.

Comment #3

Figure 1 (chemical structure of Astaxanthin) should be redrawn using a software for clarity. Astaxanthin contains a stereocenter in each of the lateral cyclohexenone rings, and consequently has three isomeric forms. For clarity, Figure 1 should also mention that the Astaxanthin structure is without the precise configuration of the alcohols.

Response

Dear Erudite Reviewer, thank you for your thoughtful comment and brilliant suggestion. We appreciate your concern in depicting the three principal astaxanthin stereoisomers. Therefore, we used the Mol View to build the new and revised Figure 1 (Page 3), which describes the three astaxanthin isomers according to a previous study published in this regard. In this figure, we also depicted the main conformational changes within the stereoisomers with highlighted text. We also updated the figure’s caption in Lines 97-110 on Pages 3-4 to depict the molecular structure of astaxanthin together with its properties. The first citation of the figure can be found in Line 79 on Page 2.

            Again, thank you for being so attentive to detail. You have an eye for improvement, and we appreciate your collaboration in reshaping our manuscript for the better. Thank you for everything!

Comment #4

L107: Systematic review?

Response

Dear Erudite Reviewer, thank you for this critical comment and insightful suggestion. We appreciate your willingness to provide us with essential feedback regarding our manuscript, which undoubtedly has been significantly improved after we adhered to all of your critical suggestions. Therefore, we improved the sentence as mentioned above, which now reads “Our current systematic review and meta-analysis aims to fill these current literature gaps.” The sentence has been significantly improved based on your critical input, and the revised sentence can be found in Lines 147-148 on Page 4 of the revised manuscript document.

            Again, thank you for your attention to detail and eye for improvement. Your suggestions have been invaluable in reshaping our manuscript for the better.

Comment #5

In Figure 3, the number of articles retrieved per database should be mentioned for clarity and transparency. It is not clear to me which automation tools were used to mark articles as ‘‘ineligible’’. In the part of ‘‘Screening’’, 6 reports were sought for retrieval and 6 reports were not retrieved. The total number of studies included in the review were 6. How does this come about from the 8 records screened?

Response

Dear Erudite Editor, thank you for your thoughtful and detailed comments. We have revised Figure 3 to include the number of articles retrieved from each database to enhance both clarity and transparency. This information is now clearly presented within the flow diagram and accompanying text. In addition, we have also specified the automation tools used during the screening process to mark articles as ineligible. This includes Covidence, as clearly stated in the revised methodology section. Regarding the “Screening” section, we acknowledge the confusion caused by the previous flow of information. We have corrected the numbers to ensure consistency throughout the diagram and text. Specifically, the transition ranges from ten records sought for retrieval to nine studies included based on eligibility after full-text review. The inconsistencies have been resolved to ensure the flow aligns logically and accurately.

            The number of studies included in the final analysis has been updated following revisions from you and the other Reviewers, which encompassed the addition of three additional studies. Therefore, the PRISMA Flow Diagram and its accompanying text have been updated to reflect the new additions.

            The new and revised Figure 3 is depicted on Page 8 of the revised manuscript document. Its first citation alongside its accompanying text is in Lines 263-257 on Page 7. Its legend is depicted in Lines 271-272 on Page 8.

We appreciate your attention to detail, which has helped us improve the overall transparency and accuracy of our reporting. Thank you for your attention to detail and eye for improvement.

Comment #6

L413-414: Scopus and Web of Science were not searched. How sure are the authors that all the relevant literature were collected?

Response

Dear Erudite Reviewer, thank you for pointing this out. We agree that including major databases such as Scopus and Web of Science is essential for ensuring comprehensive coverage of literature. In response to your comment, we conducted additional research in both Scopus and Web of Science as part of our revision process. These expanded searches led to the identification of three additional relevant articles, which have now been included in the final review. The inclusion of these studies has been reflected in the updated PRISMA flow diagram (Page 8) and accompanying text (Lines 271-272 on Page 8), as well as in the revised discussion and meta-analyses (Lines 280-571 on Pages 12-22).  The manuscript’s methodology has also been updated, as revised in Lines 253-257 on Page 7 and Lines 160-162 on Page 5 of the revised manuscript document. We believe that the incorporation of these sources has strengthened the rigor and completeness of our review, and we are confident that the revised search strategy captures the relevant literature more thoroughly. Therefore, our manuscript encompasses results from nine clinical trials now.

Thank you again for your valuable feedback, which contributed to improving the quality and comprehensiveness of our work.

Comment #7

Was the study protocol registered with any such systematic review registries like the International Prospective Register of Systematic Reviews (PROSPERO)?

Response

Dear Erudite Reviewer, thank you for your question. The review protocol was not registered in a systematic review registry such as PROSPERO. At the time of initiating the review, our primary focus was on conducting a timely and thorough synthesis of the available literature, and protocol registration was unfortunately not completed. We acknowledge that registration is recommended; however, at this time, our systematic review and meta-analysis does not possess a registration number.

            Since our results are critical and reflect the utmost criteria while evaluating recent evidence, we kindly request your approval to publish our results without a registration number at this time. We need to maintain the dissemination of critical findings like those in our meta-analysis, rapid and constant. We appreciate your understanding and look forward to your feedback and positive response.

Thank you for your attention to this matter.

Comment #8

In L505: Your conclusions and any future research directions MUST NOT have figures, reference citations or tables. Please revise.

Response

Dear Erudite Reviewer, thank you for this comment. We appreciate your willingness to provide essential feedback while improving our manuscript’s quality. We agree with you that our conclusion section must be free of figures and other content elements. Therefore, we removed all the aforementioned elements from this section. Notably, only the removal of the funnel plot analysis was necessary, and this has been done. Therefore, our manuscript’s conclusions lack figures, reference citations, or tables. The funnel plot analysis and its accompanying text are now presented in subsection “4.2. Dose-Response Meta-Analysis and Plot Asymmetry Analysis” (Lines 554-571 on Pages 20-22) of the revised manuscript document.

            Again, thank you for your thoughtful assessment of our manuscript. After addressing all your concerns, a significantly improved version emerged. We believe that our manuscript has been reshaped for the better, and we look forward to your positive response to the revised version.

Comment #9

I annotated some comments in the attached PDF file.

Response

We read your corrections in the attached PDF document. I apologize that we cannot upload print screens of your corrections during these significant revisions, as this would undoubtedly enhance the process of showing the revisions to you. We have made every effort to meet your standards, and all attributed corrections are disclosed below for your review.

Correction n° 1

This correction has been made in Lines 26-29 on Page 1. Thank you for your attention to detail and eye for improvement.

Correction n° 2

This correction has been made in Lines 37-40 on Page 1. Thank you for bringing this to our attention.

Correction n° 3

This correction has been made in Lines 51-52 on Page 2. Thank you for being so coordinated with our research.

Correction n° 4

This correction has been made in Line 131 on Page 4. Thank you for addressing this critical suggestion with us.

Correction n° 5

This correction has been made in Line 133 on Page 4. Thank you for your eye for improvement.

Correction n° 6

This correction has been made in Line 134 on Page 4. Thank you for this correction, which significantly enriched our manuscript document.

Correction n° 7

This correction has been made in Line 147 on Page 4. Thank you for bringing this to our attention.

Correction n° 8

This correction has been made in Line 148 on Page 4. Thank you for this suggestion.

Correction n° 9

This correction has been made in Lines 97-110 on Pages 3-4 and Lines 111-123 on Page 4. Thank you for this critical suggestion.

Correction n° 10

This correction has been made in Line 281 on Page 12. Thank you for bringing this to our attention.

Correction n° 11

This correction has been made in Lines 312-313 on Page 13. Thank you for being so helpful in maintaining the academic integrity of our research.

Correction n° 12

This correction has been made in Line 281 on Page 12 and Lines 312-313 on Page 13. Thank you for addressing this critical concern with us.

Correction n° 13

This correction has been made in Line 217 on Page 6. Thank you for your eye for improvement.

I, the corresponding author of the manuscript "Therapeutic Potential of Astaxanthin in Obesity: A Systematic Review and Meta-Analysis with Dose-Response Assessment" under the assigned ID pharmaceuticals-3792069, on behalf of my coauthors, once again extend my heartfelt gratitude to the knowledgeable Editor-in-Chief and reviewers for their time and expertise in revising our manuscript. After we addressed their constructive and refined feedback and suggestions, a significantly improved manuscript version emerged. Undoubtedly, their insightful suggestions and feedback have significantly enhanced the quality of our manuscript. We respectfully are at the disposal of the Editor-in-Chief and the Reviewer to address any additional suggestions regarding our publication. If you are satisfied with our newly refined and significantly improved version, we look forward to the acceptance of our article for publication in this prestigious journal, Pharmaceuticals. Thank you once again for your time and expertise.

Round 2

Reviewer 3 Report

Comments and Suggestions for Authors

Dear Author,

I am pleased to inform you that the review and revision process for your manuscript, "Therapeutic Potential of Astaxanthin in Obesity: A Systematic Review and Meta-Analysis with Dose-Response Assessment," has been completed.

As a reviewer, I would like to express my appreciation for your diligent and thorough responses to all review comments, as well as your efforts to enhance the quality of the manuscript through substantial revisions. In particular, your clarifications regarding methodology, provision of additional data, and improvements in clarity and scientific rigor have significantly elevated the quality of your work.

Accordingly, I am pleased to notify you that your manuscript has been accepted for publication. Thank you for your hard work and sincere engagement throughout the review process. I look forward to your continued contributions to the academic community.

Best regards,

Author Response

Dear Erudite Reviewer, thank you very much for your kind message and for notifying me of the acceptance of our manuscript.

I sincerely appreciate your thoughtful and constructive feedback throughout the review process. Your insights were instrumental in helping us refine and strengthen the manuscript.

Warm regards,

The Authors

Reviewer 4 Report

Comments and Suggestions for Authors

Dear authors,

Most of my suggestions were considered (and I only provided minor corrections in the PDF).

I would encourage the authors  to consider registering their review protocol with PROSPERO (https://www.crd.york.ac.uk/prospero/) or another suitable registry. Although the review has already been conducted, retrospective registration is possible with PROSPERO and is usually straightforward, and would improve the transparency and credibility of the work. Protocol registration helps prevent duplication of efforts, clarifies any deviations from the initial methodology, and aligns the study with current best practices in systematic review reporting

Author Response

RESPONSE TO REVIEWERS' COMMENTS

Manuscript number: pharmaceuticals-3792069 Pharmaceuticals (MDPI)

"Therapeutic Potential of Astaxanthin in Obesity: A Systematic Review and Meta-Analysis with Dose-Response Assessment"

The authors express their gratitude to the Editor-in-Chief and the Reviewer for their thorough evaluation of our manuscript. Their feedback has greatly enhanced our work. We have carefully implemented their suggestions, with significant revisions highlighted in YELLOW for reference, and a note for additional corrections in a different color. A detailed point-by-point response to each comment is provided below. We appreciate the time and effort invested by both, as their contributions have strengthened our manuscript.

Comment #1

I would encourage the authors to consider registering their review protocol with PROSPERO (https://www.crd.york.ac.uk/prospero/) or another suitable registry. Although the review has already been conducted, retrospective registration is possible with PROSPERO and is usually straightforward, and would improve the transparency and credibility of the work. Protocol registration helps prevent duplication of efforts, clarifies any deviations from the initial methodology, and aligns the study with current best practices in systematic review reporting

Response

Dear Reviewer, thank you for your valuable suggestion regarding protocol registration. We fully agree that registering the review protocol enhances transparency, minimizes duplication, and aligns the study with best practices in systematic review reporting. Accordingly, we will proceed with the retrospective registration of our review in PROSPERO. We appreciate your thoughtful recommendation and your commitment to improving the quality and credibility of systematic review research.

Comment #2

Dear authors, most of my suggestions were considered (and I only provided minor corrections in the PDF).

Response

We have reviewed the corrections you provided in the attached PDF document. I apologize for our inability to upload screenshots of your corrections during these significant revisions, as this would greatly facilitate the process of showing you the changes made. We have made every effort to meet your standards, and all attributed corrections are listed below for your review.

Correction n° 1

This correction has been made in Line 81 on Page 2. Thank you for bringing this to our attention.

Correction n° 2

This correction has been made in Line 82 on Page 3. Thank you for bringing this to our attention.

Correction n° 3

This correction has been made in Lines 85-86 on Page 3. Thank you for being so coordinated with our research.

Correction n° 4

This correction has been made in Line 89 on Page 3. Thank you for addressing this critical suggestion with us.

Correction n° 5

This correction has been made in Line 89 on Page 3. Thank you for addressing this critical suggestion with us.

Correction n° 6

Dear Erudite Reviewer, thank you for this important suggestion. As you might see in our manuscript’s abstract and discussion, the main point of our manuscript is to discuss the anti-obesity effects of astaxanthin in adipose tissue and myotubules because of their significance in weight management. In the paragraph you mentioned in your comment, we have already proposed fat tissue as an anti-obesity organ related to astaxanthin potential. Further, in the point of your comment, we mentioned myotubules from the skeletal muscles. Therefore, since your concern was to only mention skeletal muscles, you must stay relaxed since we’ve mentioned fat tissue earlier.

            We appreciate your attention to detail and eye for improvement, and we look forward to receiving the approval for publication of our manuscript shortly.

Correction n° 7

This correction has been made in Line 97 on Page 3. Thank you for bringing this to our attention.

Correction n° 8

Dear Erudite Reviewer, thank you for your comment. According to the reference cited, the correct form is as it is in the manuscript, as “oxi groups.” See the phrase from the reference below.

The oxygen atom can be present in OH groups such as in zeaxanthin or as oxi-groups such as in canthaxanthin, or in a combination of both, such as in ASTX.” (10.1016/j.jafr.2023.100685)

Thank you for bringing this to our attention.

Correction n° 9

This correction has been made in Line 104 on Page 3. Thank you for this critical suggestion.

Correction n° 10

This correction has been made in Lines 106-107 on Page 3. Thank you for bringing this to our attention.

Correction n° 11

This correction has been made in Line 112 on Page 4. Thank you for being so helpful in maintaining the academic integrity of our research.

Correction n° 12

This correction has been made in Line 137 on Page 4. Thank you for addressing this critical concern with us.

I, the corresponding author of the manuscript "Therapeutic Potential of Astaxanthin in Obesity: A Systematic Review and Meta-Analysis with Dose-Response Assessment" under the assigned ID pharmaceuticals-3792069, on behalf of my coauthors, once again extend my heartfelt gratitude to the knowledgeable Editor-in-Chief and reviewers for their time and expertise in revising our manuscript. After we addressed their constructive and refined feedback and suggestions, a significantly improved manuscript version emerged. Undoubtedly, their insightful suggestions and feedback have significantly enhanced the quality of our manuscript. We respectfully are at the disposal of the Editor-in-Chief and the Reviewer to address any additional suggestions regarding our publication. If you are satisfied with our newly refined and significantly improved version, we look forward to the acceptance of our article for publication in this prestigious journal, Pharmaceuticals. Thank you once again for your time and expertise.